# FAST-DIPS: Adjoint-Free Analytic Steps and Hard-Constrained Likelihood Correction for Diffusion-Prior Inverse Problems

**Minwoo Kim**[*], **Seunghyeok Shin**[*], **Hongki Lim**[†]
Department of Electrical and Computer Engineering, Inha University
Incheon, 22212, South Korea
{ququlza1520, ssh8642}@inha.edu, hklim@inha.ac.kr

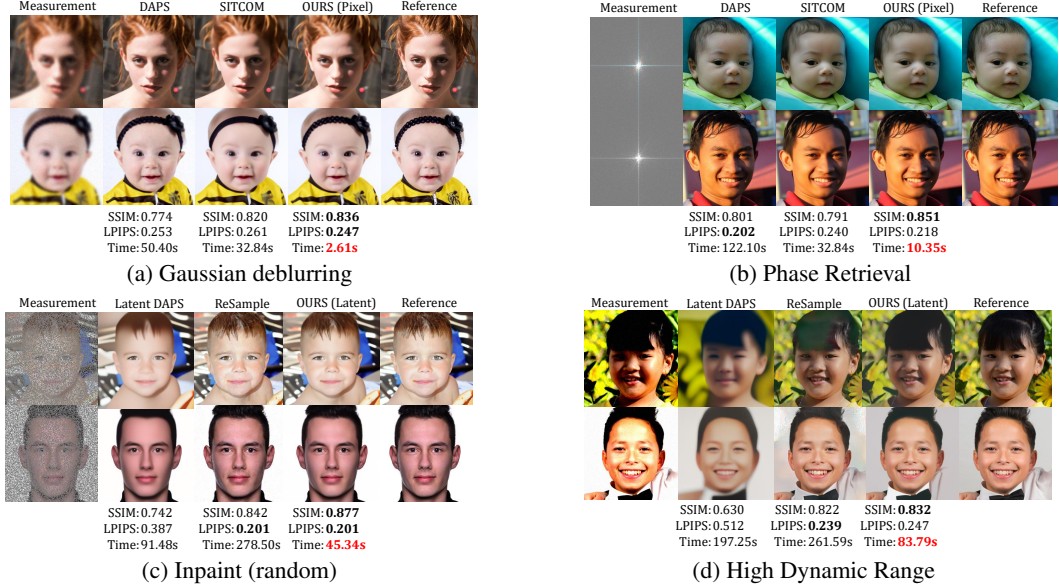

Figure 1: FFHQ results on four inverse problems: (a) Gaussian deblurring, (b) phase retrieval, (c) random inpainting, (d) HDR. Each panel shows the measurement, baselines, our FAST-DIPS output, and the reference. SSIM/LPIPS and average per-image run-time (s) are overlaid; FAST-DIPS attains comparable or higher quality with markedly lower run-time.

## ABSTRACT

Training-free diffusion priors enable inverse-problem solvers without retraining, but for nonlinear forward operators data consistency often relies on repeated derivatives or inner optimization/MCMC loops with conservative step sizes, incurring many iterations and denoiser/score evaluations. We propose a training-free solver that replaces these inner loops with a hard measurement-space feasibility constraint (closed-form projection) and an analytic, model-optimal step size, enabling a small, fixed compute budget per noise level. Anchored at the denoiser prediction, the correction is approximated via an adjoint-free, ADMM-style splitting with projection and a few steepest-descent updates, using one VJP and either one JVP or a forward-difference probe, followed by backtracking and decoupled re-annealing. We prove local model optimality and descent under backtracking for the step-size rule, and derive an explicit KL bound for mode-substitution re-annealing under a local Gaussian conditional surrogate. We also develop a latent variant and a one-parameter pixel→latent hybrid schedule. Experiments achieve competitive PSNR/SSIM/LPIPS with up to $19.5\times$ speedup, without hand-coded adjoints or inner MCMC. Code and data: 

---

[*]These authors contributed equally to this work
[†]Corresponding author

# 1 INTRODUCTION

Inverse problems seek to recover an unknown signal $\mathbf{x}$ from partial and noisy measurements $\mathbf{y} = \mathcal{A}(\mathbf{x}) + \mathbf{n}$. Such problems are ubiquitous in science and engineering, yet they are often ill-posed: distinct $\mathbf{x}$ can produce similar $\mathbf{y}$ due to the structure of the operator $\mathcal{A}$ and measurement noise $\mathbf{n}$. The Bayesian viewpoint constrains the solution via a prior and asks to sample from $p(\mathbf{x} \mid \mathbf{y}) \propto p(\mathbf{y} \mid \mathbf{x})\, p(\mathbf{x})$.

Diffusion models have emerged as a powerful class of learned priors for modeling complex data distributions, including natural images (Ho et al. (2020); Song & Ermon (2020); Song et al. (2021a;b); Dhariwal & Nichol (2021); Karras et al. (2022); Song et al. (2023); Lu & Song (2025)). Through reverse-time dynamics, they progressively transform simple noise into samples from the target distribution. This generative mechanism offers a natural framework for inverse problems, where the reverse-time SDE is guided by measurements to draw posterior samples.

Diffusion-based inverse problem solvers span both task-specific conditional models and training-free posterior samplers. Task-specific conditional diffusion/bridge models are trained for particular restoration or image-to-image tasks (Saharia et al. (2022); Liu et al. (2023)). In contrast, many training-free solvers start from an unconditional pretrained prior and impose data consistency at sampling time, including inference-time inpainting (Lugmayr et al. (2022)), linear-operator frameworks (Kawar et al. (2022); Wang et al. (2023)), and decoupled/posterior-aware updates (Chung et al. (2023a;b); Dou & Song (2024); Zhang et al. (2025)). Other lines formulate plug-and-play optimization with diffusion denoisers (Zhu et al. (2023); Rout et al. (2024); Wu et al. (2024); Xu & Chi (2024); Mardani et al. (2024); Wang et al. (2024); Zheng et al. (2025)), Monte-Carlo guidance (Cardoso et al. (2024)), or aim for faster sampling via preconditioning, parallelization, or fast/few-step samplers (Garber & Tirer (2024); Cao et al. (2024); Liu et al. (2024); Chung et al. (2024)).

A central practical question is how data consistency is enforced. Many training-free designs inject measurements through an operator-aware data-consistency (likelihood) update (Kawar et al. (2022); Wang et al. (2023); Rout et al. (2023); Liu et al. (2024); Pandey et al. (2024); Cao et al. (2024); Garber & Tirer (2024); Dou & Song (2024); Cardoso et al. (2024); Chung et al. (2024)). For some linear degradations, this update admits efficient closed-form structure by exploiting explicit adjoints and/or pseudo-inverses (Kawar et al. (2022); Wang et al. (2023)). When such closed-form operator primitives are unavailable (e.g., complex, ill-conditioned, or simulator-based forward models implemented in autodiff frameworks but lacking closed-form adjoints/pseudo-inverses), the likelihood step is often implemented via inner iterative solvers (e.g., multiple gradient steps on a data-fidelity objective) or MCMC/Langevin subloops with conservative step sizes, which increases wall-clock cost due to many inner iterations and repeated score/denoiser calls (Zhu et al. (2023); Wu et al. (2024); Xu & Chi (2024); Mardani et al. (2024); Wang et al. (2024); Zhang et al. (2025)).

A complementary design axis is latent vs. pixel execution. Latent diffusion models reduce dimensionality and sampling cost, and many recent posterior samplers therefore operate in latent space (Song et al. (2024); Rout et al. (2024); Zhang et al. (2025)), with an encoder $\mathcal{E}$ and decoder $\mathcal{D}$. When fidelity is defined in pixel space, latent-space likelihood updates require backpropagating through the decoder, e.g., $\nabla_{\mathbf{z}}\|\mathcal{A}(\mathcal{D}(\mathbf{z})) - \mathbf{y}\|^2$, which can be a throughput bottleneck. A pixel-space correction instead decodes $\mathbf{z} \to \mathbf{x}$, applies a data-fidelity correction in image space, then re-encodes $\mathbf{x} \to \mathbf{z}$; this avoids decoder backprop during correction, but can be sensitive to how Jacobian–vector products (JVPs) are computed for highly nonlinear $\mathcal{A}$.

Taken together, these considerations motivate training-free solvers that (i) impose data consistency without operator-specific primitives (e.g., hand-coded adjoints, pseudo-inverses, or SVDs), (ii) keep the per-noise-level correction lightweight by using a small, fixed compute budget (few inner steps and few operator/gradient calls) while delivering competitive reconstruction quality, and (iii) balance pixel- and latent-space computation to trade early-time throughput for late-time fidelity. We propose FAST-DIPS (Fast And STable Diffusion-prior Inverse Problem Solver), which centers each noise-level update at the denoiser prediction and applies a measurement-space feasibility correction with an interpretable residual budget under additive white Gaussian noise (AWGN), followed by decoupled re-annealing. Our implementation is adjoint-free and uses only a small fixed number of inexpensive autodiff primitives per level, avoiding hand-coded adjoints and inner MCMC while remaining applicable to nonlinear forward operators. We also develop a latent counterpart and a

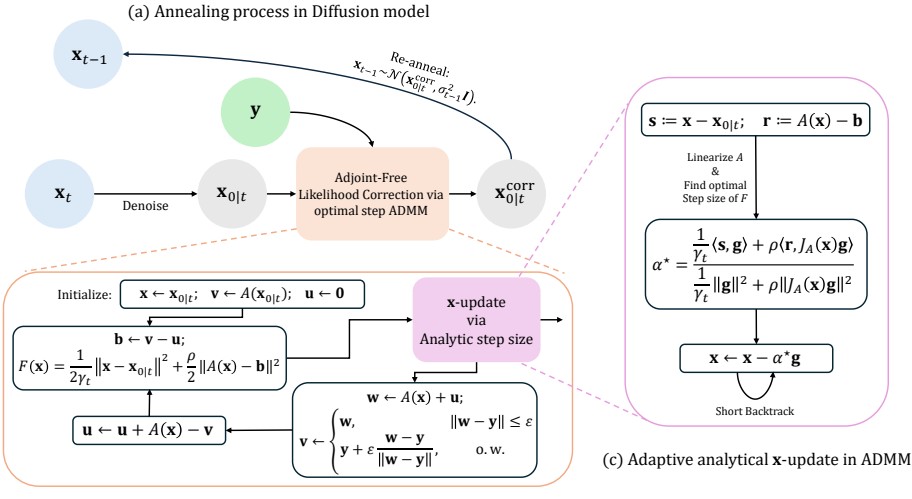

Figure 2: **FAST-DIPS method sketch.** At each noise level $t$, we (1) compute a denoiser anchor $\mathbf{x}_{0|t}$, (2) perform a hard-constrained measurement-space correction under an $\ell_2$ feasibility ball via a few-step ADMM-style splitting (closed-form projection + a few descent steps with analytic $\alpha^\star$ using one VJP and either one JVP or a forward-difference probe), and (3) re-anneal to obtain $\mathbf{x}_{t-1}$.

one-switch pixel→latent hybrid schedule that corrects in pixels early (cheap) and in latents late (manifold-faithful).

FAST-DIPS differs from PnP–ADMM (Chan et al. (2016); Venkatakrishnan et al. (2013)) in that the denoiser is not a (surrogate) proximal operator; it only provides an anchor, while an ADMM-style splitting enforces an explicit hard measurement-space constraint via projection. Unlike quadratic penalties $\lambda\|\mathcal{A}(\mathbf{x}) - \mathbf{y}\|^2$ that require tuning and can be noise-sensitive, we use an indicator confidence-set likelihood with an interpretable residual budget. Unlike coupled DPS-style guidance (Chung et al. (2023a)), we keep traversal decoupled (Zhang et al. (2025)) and apply diffusion-kernel transport after each correction via mode-substitution re-annealing. Our pixel→latent hybrid reduces decoder-Jacobian cost early while preserving manifold fidelity late. Finally, while we use autodiff for VJPs (and one JVP or a forward-difference surrogate) (Baydin et al. (2018)), we avoid hand-crafted adjoints and closed-form Jacobians/pseudo-inverses of $\mathcal{A}$.

Our contributions can be summarized as follows:

- **Adjoint-free feasibility correction.** At each diffusion step, we enforce measurement-space feasibility around the denoiser prediction with a hard residual budget and a schedule-aware trust region, using an adjoint-free splitting solver with a closed-form projection and a small fixed inner compute budget, avoiding hand-crafted adjoints and inner MCMC.

- **Theory for step size and re-annealing.** We prove the proposed step size is locally model-optimal and, with backtracking, yields descent for the current subproblem (Proposition 1), and we provide an explicit KL bound for the per-step Gaussian-injected distribution induced by mode-substitution re-annealing under a local Gaussian conditional surrogate (Proposition 6).

- **Latent and hybrid execution with speed gains.** We derive a latent counterpart via $\mathcal{A} \circ \mathcal{D}$ and a one-switch pixel→latent hybrid that balances early-time efficiency and late-time fidelity; across eight linear and nonlinear tasks, we achieve competitive quality and up to $19.5\times$ speedup on FFHQ pixel-space tasks with robust default hyperparameters.

Orthogonal to our contributions, fast/few-step samplers and parallel sampling schemes can reduce the diffusion sampling cost (Zhao et al. (2024); Cao et al. (2024); Liu et al. (2024); Chung et al. (2024)). FAST-DIPS complements such advances by minimizing inner correction cost and adjoint engineering while preserving an explicit measurement-space feasibility set via projection in the split variable, so these techniques are composable with our approach.

## 2 METHOD

### 2.1 HIGH-LEVEL OVERVIEW AND DESIGN CHOICES

We briefly summarize the goals and main design choices of FAST-DIPS before introducing the detailed derivations.

**Setting and practical issues.** We use a pretrained diffusion model as a prior for inverse problems $\mathbf{y} = \mathcal{A}(\mathbf{x}_0) + \mathbf{n}$, where $\mathcal{A}$ may be nonlinear. Existing training-free solvers often enforce data consistency via many gradient steps or inner Langevin/MCMC chains at each noise level, and control their influence through a soft data-fidelity weight. This can be computationally heavy (many evaluations of $\mathcal{A}$ and its gradient) and sensitive to step sizes and noise miscalibration.

**Per-level update in FAST-DIPS.** At each diffusion level $t$, FAST-DIPS performs a single correct–then–noise update with three ingredients:

1. **Local hard-constrained MAP.** The denoiser output $\mathbf{x}_{0|t}$ is treated as the center of a local Gaussian prior, and data consistency is enforced via a simple measurement-space constraint $\|\mathcal{A}(\mathbf{x}) - \mathbf{y}\| \leq \varepsilon$. This defines a constrained proximal problem whose solution $\mathbf{x}_{0|t}^{\text{corr}}$ is the likelihood correction at level $t$.

2. **Adjoint-free ADMM with analytic step size.** We solve the constrained problem with a small, fixed number of deterministic ADMM-style iterations. The split measurement variable is updated by a closed-form projection onto the constraint set, while the image variable is updated by a few steepest-descent steps whose step size is computed per step by minimizing a local 1D quadratic model (and refined by backtracking). This keeps the number of evaluations of $\mathcal{A}$ and its derivatives small. No inner MCMC chain and no hand-coded adjoint operator are required: VJPs/JVPs are obtained from autodiff, and when JVPs are unavailable we use a single finite-difference probe (Remark 4). This leads to substantially fewer total function calls than methods based on long gradient/MCMC inner loops.

3. **Re-annealing.** After the correction, we re-anneal by adding Gaussian noise with the next diffusion variance, decoupling the measurement-aware update from the diffusion noise.

**Theoretical support.** The inner $\mathbf{x}$-update uses an analytic step size that minimizes the local quadratic model, and Armijo backtracking ensures an accepted step decreases the current $\mathbf{x}$-subproblem objective under mild regularity (Proposition 1; proof in Appendix A.3). For completeness, Appendix A.3 also includes (i) a fixed-point $\Rightarrow$ KKT characterization for the exact scaled-ADMM mapping (Proposition 5), and (ii) a KL bound for the Gaussian-injected distribution induced by mode-substitution re-annealing under a local Gaussian conditional approximation (Proposition 6). With nonlinear $\mathcal{A}$ the per-level problem is generally nonconvex, and we do not claim global convergence.

**Practical per-step recipe.** Figure 2 and Algorithms 1–2 give the full procedure. Let $\mathbf{x}_{\text{den}}$ denote the pretrained denoiser. One reverse step of FAST-DIPS at level $t$ proceeds as: (i) *Anchor and initialization:* compute the denoiser anchor $\mathbf{x}_{0|t} = \mathbf{x}_{\text{den}}(\mathbf{x}_t, \sigma_t)$ and set $\mathbf{x}^{(0)} \leftarrow \mathbf{x}_{0|t}$, $\mathbf{v}^{(0)} \leftarrow \mathcal{A}(\mathbf{x}^{(0)})$, $\mathbf{u}^{(0)} \leftarrow \mathbf{0}$. (ii) *Few-step ADMM-style correction:* for $k = 0, \ldots, K-1$, update $\mathbf{x}$ by applying a few successive steepest-descent steps on $F$ in Equation 17 using the gradient in Equation 18, with the step size computed as $\alpha^\star$ in Equation 21 (via one VJP and one JVP, or the finite-difference probe in Equation 22/Remark 4) and then refined by backtracking. Then update $\mathbf{v}^{(k+1)}$ by the projection in Equation 14–Equation 16 and $\mathbf{u}^{(k+1)}$ by Equation 15. After $K$ iterations, set $\mathbf{x}_{0|t}^{\text{corr}} \leftarrow \mathbf{x}^{(K)}$. (iii) *Re-anneal:* sample $\mathbf{x}_{t-1} = \mathbf{x}_{0|t}^{\text{corr}} + \sigma_{t-1}\boldsymbol{\xi}$ with $\boldsymbol{\xi} \sim \mathcal{N}(\mathbf{0}, I)$.

### 2.2 PROBLEM SETUP

Let $\mathbf{x}_0 \in \mathbb{R}^{C \times H \times W}$ denote the clean image stacked as a vector and

$$\mathbf{y} = \mathcal{A}(\mathbf{x}_0) + \mathbf{n}, \qquad \mathbf{n} \sim \mathcal{N}(\mathbf{0}, \beta^2 I), \tag{1}$$

where $\mathcal{A} : \mathbb{R}^{CHW} \rightarrow \mathbb{R}^m$ is a (possibly nonlinear) forward operator. Throughout the paper we assume additive white Gaussian noise (AWGN) with variance $\beta^2$ and use the standard Euclidean norm in measurement space.

## 2.3 PROBABILISTIC MOTIVATION AND THE PER-LEVEL OBJECTIVE

The reverse process of the diffusion model, conditioned on $\mathbf{y}$, is described by the reverse-time SDE (Song et al. (2021b)):

$$d\mathbf{x}_t = -2\dot{\sigma}(t)\sigma(t)\nabla_{\mathbf{x}_t} \log p(\mathbf{x}_t|\mathbf{y}; \sigma_t)\, dt + \sqrt{2\dot{\sigma}(t)\sigma(t)}\, d\mathbf{w}_t. \qquad (2)$$

We do not use the SDE form directly; instead, we derive a fast per-level update from the conditional factorization in Equation 3. At each diffusion level $t$ we maintain a state $\mathbf{x}_t$ and wish to transform the time–marginal $p(\mathbf{x}_t \mid \mathbf{y})$ into a good approximation to $p(\mathbf{x}_{t-1} \mid \mathbf{y})$ by performing a local, measurement-aware likelihood correction around the denoiser's prediction. The derivation proceeds from the conditional factorization

$$p(\mathbf{x}_0 \mid \mathbf{x}_t, \mathbf{y}) \propto p(\mathbf{x}_0 \mid \mathbf{x}_t)\, p(\mathbf{y} \mid \mathbf{x}_0), \qquad (3)$$

and two modeling choices: a local Laplace surrogate for $p(\mathbf{x}_0 \mid \mathbf{x}_t)$ and a set-valued likelihood in measurement space.

**Local prior surrogate around the denoiser.** Write

$$\mathbf{x}_{0|t} := \mathbf{x}_{\mathrm{den}}(\mathbf{x}_t, \sigma_t), \qquad (4)$$

and approximate the intractable $p(\mathbf{x}_0 \mid \mathbf{x}_t)$ by a Gaussian centered at $\mathbf{x}_{0|t}$,

$$p(\mathbf{x}_0 \mid \mathbf{x}_t) \approx \tilde{p}_t(\mathbf{x}_0 \mid \mathbf{x}_t) \propto \exp\left( -\tfrac{1}{2\gamma_t} \|\mathbf{x}_0 - \mathbf{x}_{0|t}\|^2 \right), \qquad (5)$$

where $\gamma_t > 0$ plays the role of a local prior variance. Inspired by schedule-tied noise-annealing heuristics in decoupled inverse solvers (Zhang et al. (2025)), we set $\gamma_t = \sigma_t^2$, so that the proximal trust region naturally tightens with annealing.

**Conservative likelihood via a measurement-space feasibility ball.** Under AWGN, the Gaussian likelihood is

$$p(\mathbf{y} \mid \mathbf{x}_0, \beta) \propto \beta^{-m} \exp\left( -\tfrac{1}{2\beta^2} \|\mathcal{A}(\mathbf{x}_0) - \mathbf{y}\|^2 \right). \qquad (6)$$

We replace $p(\mathbf{y} \mid \mathbf{x}_0, \beta)$ by a set-valued surrogate likelihood $\tilde{\ell}_\varepsilon(\mathbf{y} \mid \mathbf{x}_0)$ to improve robustness to noise miscalibration. If $\beta$ is known, then for any confidence level $1 - \delta$, a $(1 - \delta)$ confidence region for the residual $\mathcal{A}(\mathbf{x}_0) - \mathbf{y}$ under Equation 6 is the Euclidean ball $\{\mathbf{v} : \|\mathbf{v} - \mathbf{y}\| \leq \varepsilon\}$ with $\varepsilon = \beta\sqrt{\chi^2_{m,1-\delta}}$ (Casella & Berger (1990)); using this region yields an indicator likelihood. In our implementation, we treat $\varepsilon$ as a user-chosen data-consistency tolerance and do not calibrate it via a specific choice of $\delta$. If $\beta$ is unknown, profiling it out gives $-\log p(\mathbf{y} \mid \mathbf{x}_0, \hat{\beta}(\mathbf{x}_0)) \propto m \log \|\mathcal{A}(\mathbf{x}_0) - \mathbf{y}\|$ (Casella & Berger (1990)), which is monotone in the residual norm and motivates enforcing $\|\mathcal{A}(\mathbf{x}_0) - \mathbf{y}\| \leq \varepsilon$ for a chosen budget $\varepsilon > 0$ (Engl & Hanke (1996)). Both viewpoints lead to

$$\tilde{\ell}_\varepsilon(\mathbf{y} \mid \mathbf{x}_0) \propto \mathbf{1}\{\|\mathcal{A}(\mathbf{x}_0) - \mathbf{y}\| \leq \varepsilon\}, \qquad (7)$$

which replaces $p(\mathbf{y} \mid \mathbf{x}_0, \beta)$ in Equation 3 and yields the hard constraint in Equation 9.

**Per-level surrogate conditional and MAP.** Combining Equation 5 and Equation 7 with Equation 3 yields

$$\tilde{p}_t(\mathbf{x}_0 \mid \mathbf{x}_t, \mathbf{y}) \propto \exp\left( -\tfrac{1}{2\gamma_t} \|\mathbf{x}_0 - \mathbf{x}_{0|t}\|^2 \right) \mathbf{1}\{\|\mathcal{A}(\mathbf{x}_0) - \mathbf{y}\| \leq \varepsilon\}. \qquad (8)$$

We take the mode of Equation 8 as the likelihood correction at level $t$, which solves

$$\mathbf{x}_{0|t}^{\mathrm{corr}} \in \arg\min_{\mathbf{x} \in \mathbb{R}^{CHW}} \frac{1}{2\gamma_t} \|\mathbf{x} - \mathbf{x}_{0|t}\|^2 \quad \text{s.t.} \quad \|\mathcal{A}(\mathbf{x}) - \mathbf{y}\| \leq \varepsilon. \qquad (9)$$

Problem Equation 9 is a hard-constrained proximal objective: the first term is a schedule-aware trust region around the denoiser estimate, while the constraint enforces measurement feasibility within a user-specified residual budget in measurement space.

## 2.4 DECOUPLED RE-ANNEALING AND CONNECTION TO TIME–MARGINALS

Let $\kappa_{t\to t-1}(\mathbf{x}_{t-1} \mid \mathbf{x}_0) = \mathcal{N}(\mathbf{x}_{t-1}; \mathbf{x}_0, \sigma_{t-1}^2 I)$ denote the diffusion kernel that transports the clean image to the next diffusion state. The exact time–marginal recursion (Zhang et al. (2025)) is

$$p(\mathbf{x}_{t-1} \mid \mathbf{y}) = \int \left[ \int \kappa_{t\to t-1}(\mathbf{x}_{t-1} \mid \mathbf{x}_0)\, p(\mathbf{x}_0 \mid \mathbf{x}_t, \mathbf{y})\, d\mathbf{x}_0 \right] p(\mathbf{x}_t \mid \mathbf{y})\, d\mathbf{x}_t. \tag{10}$$

Thus, transforming $p(\mathbf{x}_t \mid \mathbf{y})$ to $p(\mathbf{x}_{t-1} \mid \mathbf{y})$ amounts to obtaining a representative $\mathbf{x}_0 \sim p(\mathbf{x}_0 \mid \mathbf{x}_t, \mathbf{y})$ and injecting Gaussian noise of variance $\sigma_{t-1}^2$. We approximate $p(\mathbf{x}_0 \mid \mathbf{x}_t, \mathbf{y})$ by $\tilde{p}_t$ in Equation 8 and substitute its mode, yielding

$$\mathbf{x}_{t-1} \;=\; \mathbf{x}_{0|t}^{\text{corr}} + \sigma_{t-1}\boldsymbol{\xi}, \qquad \boldsymbol{\xi} \sim \mathcal{N}(\mathbf{0}, I). \tag{11}$$

Appendix A.3 provides an explicit KL bound for the mode-substitution re-annealing step (for fixed $\mathbf{x}_t$), comparing it to the per-step Gaussian-injected distribution obtained by sampling from a local Gaussian surrogate for $p(\mathbf{x}_0 \mid \mathbf{x}_t, \mathbf{y})$ (Proposition 6).

## 2.5 PIXEL-SPACE ADMM SOLVER WITH ADJOINT-FREE UPDATES

We solve Equation 9 via variable splitting (Combettes & Pesquet (2011); Boyd et al. (2011)) in pixel space. Introduce an auxiliary $\mathbf{v} \approx \mathcal{A}(\mathbf{x})$ and the feasibility set $\mathcal{C} := \{\mathbf{v} : \|\mathbf{v} - \mathbf{y}\| \le \varepsilon\}$. Consider

$$\min_{\mathbf{x},\mathbf{v}} \;\; \underbrace{\frac{1}{2\gamma_t}\|\mathbf{x} - \mathbf{x}_{0|t}\|^2}_{f(\mathbf{x})} + \underbrace{\iota_{\mathcal{C}}(\mathbf{v})}_{g(\mathbf{v})} \quad \text{s.t.} \quad \mathcal{A}(\mathbf{x}) - \mathbf{v} = \mathbf{0}, \tag{12}$$

where $\iota_{\mathcal{C}}$ is the indicator of $\mathcal{C}$. Using a scaled ADMM-style augmented Lagrangian splitting with penalty $\rho > 0$ and scaled dual $\mathbf{u}$, we iterate

$$\mathbf{x}^{k+1} = \arg\min_{\mathbf{x}} \; \frac{1}{2\gamma_t}\|\mathbf{x} - \mathbf{x}_{0|t}\|^2 + \frac{\rho}{2}\|\mathcal{A}(\mathbf{x}) - \mathbf{v}^k + \mathbf{u}^k\|^2, \tag{13}$$

$$\mathbf{v}^{k+1} = \Pi_{\mathcal{C}}\Big(\mathcal{A}(\mathbf{x}^{k+1}) + \mathbf{u}^k\Big), \tag{14}$$

$$\mathbf{u}^{k+1} = \mathbf{u}^k + \mathcal{A}(\mathbf{x}^{k+1}) - \mathbf{v}^{k+1}. \tag{15}$$

Let $\mathbf{b}^k := \mathbf{v}^k - \mathbf{u}^k$ for brevity. *In* FAST-DIPS *we run only a small fixed K and do not solve Equation 13 exactly: we approximate the* $\mathbf{x}$*-update by a steepest-descent step on F with analytic step size and backtracking (below).* Each correction iteration requires at least one forward evaluation of $\mathcal{A}$ (often reused across forming $\mathcal{A}(\mathbf{x}) - \mathbf{b}^k$, the $\mathbf{v}$-update, and the $\mathbf{u}$-update by caching), one VJP to form $J_{\mathcal{A}}(\mathbf{x})^\top(\mathcal{A}(\mathbf{x}) - \mathbf{b}^k)$, and either one JVP or one additional forward probe for $J_{\mathcal{A}}(\mathbf{x})\mathbf{g}$; backtracking may add a few extra forward calls. The $\mathbf{v}$-update is closed-form.

**Closed-form projection onto the measurement ball.** For $\mathcal{C} = \{\mathbf{v} \in \mathbb{R}^m : \|\mathbf{v} - \mathbf{y}\| \le \varepsilon\}$, the Euclidean projection in Equation 14 is the standard radial shrink (Parikh & Boyd (2014))

$$\Pi_{\mathcal{C}}(\mathbf{w}) \;=\; \begin{cases} \mathbf{w}, & \|\mathbf{w} - \mathbf{y}\| \le \varepsilon, \\[2mm] \mathbf{y} + \varepsilon\, \dfrac{\mathbf{w} - \mathbf{y}}{\|\mathbf{w} - \mathbf{y}\|}, & \text{otherwise.} \end{cases} \tag{16}$$

**Efficient x-update.** Define the smooth objective for Equation 13

$$F(\mathbf{x}) \;=\; \frac{1}{2\gamma_t}\|\mathbf{x} - \mathbf{x}_{0|t}\|^2 + \frac{\rho}{2}\|\mathcal{A}(\mathbf{x}) - \mathbf{b}^k\|^2. \tag{17}$$

Let $\mathbf{s} := \mathbf{x} - \mathbf{x}_{0|t}$ and $\mathbf{r} := \mathcal{A}(\mathbf{x}) - \mathbf{b}^k$. Its gradient is

$$\mathbf{g} \;=\; \nabla F(\mathbf{x}) \;=\; \frac{1}{\gamma_t}(\mathbf{x} - \mathbf{x}_{0|t}) \;+\; \rho\, J_{\mathcal{A}}(\mathbf{x})^\top\big(\mathcal{A}(\mathbf{x}) - \mathbf{b}^k\big), \qquad \mathbf{x} \leftarrow \mathbf{x} - \alpha\,\mathbf{g}, \tag{18}$$

where $J_{\mathcal{A}}(\mathbf{x})$ is the Jacobian of $\mathcal{A}$ at $\mathbf{x}$. Crucially, both the VJP $J_{\mathcal{A}}(\mathbf{x})^\top\mathbf{r}$ and the JVP $J_{\mathcal{A}}(\mathbf{x})\mathbf{g}$ can be obtained from autodiff (Baydin et al. (2018)).

We linearize $\mathcal{A}$ along the descent direction:

$$\mathcal{A}(\mathbf{x} - \alpha\mathbf{g}) \approx \mathcal{A}(\mathbf{x}) - \alpha J_{\mathcal{A}}(\mathbf{x})\mathbf{g}. \tag{19}$$

Substituting Equation 19 into $F(\mathbf{x} - \alpha\mathbf{g})$ yields a one-dimensional quadratic model (Nocedal & Wright (2006))

$$\tilde{F}(\alpha) = \frac{1}{2\gamma_t}\|\mathbf{s} - \alpha\mathbf{g}\|^2 + \frac{\rho}{2}\|\mathbf{r} - \alpha J_{\mathcal{A}}(\mathbf{x})\mathbf{g}\|^2, \tag{20}$$

whose exact minimizer is

$$\alpha^\star = \frac{\frac{1}{\gamma_t}\langle \mathbf{s}, \mathbf{g}\rangle + \rho\langle \mathbf{r}, J_{\mathcal{A}}(\mathbf{x})\mathbf{g}\rangle}{\frac{1}{\gamma_t}\|\mathbf{g}\|^2 + \rho\|J_{\mathcal{A}}(\mathbf{x})\mathbf{g}\|^2}. \tag{21}$$

The numerator can be rewritten as $\frac{1}{\gamma_t}\langle \mathbf{s}, \mathbf{g}\rangle + \rho\langle \mathbf{r}, J_{\mathcal{A}}(\mathbf{x})\mathbf{g}\rangle = \langle \frac{1}{\gamma_t}\mathbf{s} + \rho J_{\mathcal{A}}(\mathbf{x})^\top\mathbf{r}, \mathbf{g}\rangle = \|\mathbf{g}\|^2$. Hence $\alpha^\star \geq 0$ whenever $\mathbf{g} \neq \mathbf{0}$ (the clamp is a numerical safeguard); if $\mathbf{g} = \mathbf{0}$ we are already stationary for the current $\mathbf{x}$-subproblem and set $\alpha = 0$. Computing $\alpha^\star$ (when $\mathbf{g} \neq \mathbf{0}$) requires one forward evaluation of $\mathcal{A}$ (to form $\mathbf{r}$), one VJP to form $\mathbf{g}$, and one JVP (or one additional forward probe) to form $J_{\mathcal{A}}(\mathbf{x})\mathbf{g}$; backtracking evaluates $F$ a few times but reuses the same descent direction. In practice we initialize $\alpha$ by $\alpha^\star$ and apply Armijo backtracking to ensure decrease of $F$.

**Proposition 1** (Local model-optimal step and descent). *Under $C^1$ regularity of $\mathcal{A}$ near $\mathbf{x}$ and local Lipschitzness of $J_{\mathcal{A}}$, if $\mathbf{g} \neq \mathbf{0}$ then $\alpha^\star$ in Equation 21 minimizes the quadratic model $\tilde{F}$ in Equation 20. Moreover, using $\alpha^\star$ to initialize an Armijo backtracking line search guarantees that the accepted step yields monotone decrease of the current $\mathbf{x}$-subproblem objective $F$ in Equation 17 (with $\mathbf{b}^k$ fixed), even when Equation 19 is only locally accurate. If $\mathbf{g} = \mathbf{0}$, then $\mathbf{x}$ is stationary for $F$ and no step is taken.*

**Step size via finite-difference JVP (fallback).**  The analytic step size $\alpha^\star$ in Equation 21 requires a JVP, $J_{\mathcal{A}}(\mathbf{x})\mathbf{g}$. When forward-mode JVPs are unavailable or impractical, we can estimate the JVP by a single forward probe (Nocedal & Wright (2006)):

$$J_{\mathcal{A}}(\mathbf{x})\mathbf{g} \approx \frac{\mathcal{A}(\mathbf{x} + \eta\mathbf{g}) - \mathcal{A}(\mathbf{x})}{\eta} =: \frac{\Delta\mathcal{A}}{\eta} \tag{22}$$

where $\eta > 0$ is a small scalar perturbation parameter. Appendix A.3 provides a stabilized closed form (Remark 4, Equation 38).

## 2.6 OPTIMALITY CONDITIONS AND FEASIBILITY

**Remark 1** (Nonconvexity and scope). *With nonlinear $\mathcal{A}$, problem Equation 9 is generally nonconvex; we do not claim global convergence. Our guarantees are local: the $\mathbf{x}$-update descends $F$ (Proposition 1 and Remark 4). Feasibility is enforced in the split measurement variable $\mathbf{v}$ via the projection Equation 16 (so $\mathbf{v}^k \in \mathcal{C}$), while the ADMM coupling encourages $\mathcal{A}(\mathbf{x}^k) \approx \mathbf{v}^k$. For completeness, Appendix A.3 provides (i) a fixed-point $\Rightarrow$ KKT characterization for the exact scaled-ADMM mapping (Proposition 5), and (ii) a KL bound that formalizes mode-substitution in re-annealing under a local Gaussian approximation (Proposition 6).*

## 2.7 LATENT FAST-DIPS

We extend FAST-DIPS to latent diffusion models (Rombach et al. (2022); Podell et al. (2024)) by applying the same per-level construction in latent space: we replace $\mathcal{A}$ by the composite operator $\mathcal{A} \circ \mathcal{D}$ (with $\mathcal{D}$ the pretrained decoder) and optimize over the latent variable. The ADMM splitting, measurement-space projection, and analytic step-size carry over via the chain rule and autodiff through $\mathcal{D}$ and $\mathcal{A}$. To balance cost and fidelity, we use a hybrid schedule: early steps (large $\sigma_t$) apply cheaper pixel-space corrections, then we switch to latent corrections once $\sigma_t \leq \sigma_{\text{switch}}$ to better conform to the learned manifold. Details appear in Appendix A.1.

# 3 EXPERIMENTS

## 3.1 EXPERIMENTAL SETUP

Our experimental setup, including the suite of inverse problems and noise levels, largely follows that of DAPS (Zhang et al. (2025)). We evaluate our method across eight tasks—five linear and three nonlinear—to demonstrate its versatility.

Table 1: Quantitative evaluation on 100 FFHQ images and 100 ImageNet images for eight inverse problems (five linear and three nonlinear). The best and second-best results within each task type (Pixel and Latent) are indicated in **bold** and underlined, respectively. Method names shown in gray denote methods designed for noiseless settings

| Task | Type | Method | FFHQ | | | | ImageNet | | | |
|---|---|---|---|---|---|---|---|---|---|---|
| | | | PSNR (↑) | SSIM (↑) | LPIPS (↓) | Run-time (s) | PSNR (↑) | SSIM (↑) | LPIPS (↓) | Run-time (s) |
| Super resolution 4× | Pixel | DAPS | 28.774 | 0.774 | 0.257 | 40.229 | 25.686 | 0.651 | 0.364 | 97.192 |
| | | SITCOM | 29.555 | **0.841** | **0.237** | 21.591 | **26.519** | **0.716** | **0.309** | 65.657 |
| | | C-IIGDM | 27.794 | 0.807 | 0.209 | 1.404 | 23.645 | 0.631 | 0.313 | 4.085 |
| | | HRDIS | 30.455 | 0.867 | 0.156 | 2.274 | 26.764 | 0.744 | 0.291 | 6.216 |
| | | Ours | **29.573** | **0.841** | 0.244 | 2.726 | 26.367 | 0.714 | 0.334 | 6.266 |
| | Latent | LatentDAPS | **29.184** | **0.825** | **0.273** | 93.383 | 26.189 | 0.702 | 0.388 | 95.675 |
| | | PSLD | 23.749 | 0.601 | 0.347 | 92.799 | 21.262 | 0.405 | 0.501 | 149.29 |
| | | ReSample | 23.317 | 0.456 | 0.507 | 248.865 | 22.152 | 0.423 | 0.470 | 275.999 |
| | | Ours | 28.634 | 0.797 | 0.283 | 45.304 | 26.298 | 0.704 | 0.377 | 46.516 |
| Inpaint (box) | Pixel | DAPS | 24.546 | 0.754 | 0.218 | 33.108 | **21.399** | 0.726 | 0.271 | 81.166 |
| | | SITCOM | 25.336 | **0.858** | **0.169** | 24.994 | 20.638 | **0.794** | **0.209** | 73.986 |
| | | C-IIGDM | 18.294 | 0.731 | 0.358 | 1.277 | 17.514 | 0.676 | 0.360 | 3.683 |
| | | HRDIS | 21.735 | 0.785 | 0.194 | 3.726 | 20.507 | 0.707 | 0.280 | 10.107 |
| | | Ours | 24.605 | 0.850 | 0.190 | 2.937 | 21.381 | 0.777 | 0.278 | 6.347 |
| | Latent | LatentDAPS | 23.530 | 0.742 | 0.369 | 91.687 | 19.630 | 0.588 | 0.522 | 96.110 |
| | | PSLD | 21.428 | 0.823 | 0.126 | 91.189 | 21.084 | 0.803 | 0.186 | 146.644 |
| | | ReSample | 19.978 | 0.796 | 0.247 | 253.162 | 18.087 | 0.713 | 0.309 | 281.831 |
| | | Ours | 24.048 | 0.829 | 0.247 | 45.276 | 19.349 | 0.716 | 0.389 | 45.989 |
| Inpaint (random) | Pixel | DAPS | 30.280 | 0.797 | 0.211 | 35.361 | 27.677 | 0.736 | 0.240 | 82.617 |
| | | SITCOM | **32.580** | **0.911** | **0.148** | 35.499 | **29.726** | **0.853** | **0.168** | 106.182 |
| | | C-IIGDM | 25.888 | 0.728 | 0.283 | 1.281 | 18.253 | 0.453 | 0.452 | 3.613 |
| | | HRDIS | 28.722 | 0.823 | 0.202 | 4.518 | 24.614 | 0.676 | 0.321 | 9.703 |
| | | Ours | 31.022 | 0.879 | 0.202 | 2.908 | 28.353 | 0.791 | 0.249 | 5.857 |
| | Latent | LatentDAPS | 25.979 | 0.742 | 0.387 | 91.480 | 22.695 | 0.567 | 0.549 | 95.442 |
| | | PSLD | 22.836 | 0.472 | 0.467 | 87.157 | 22.761 | 0.522 | 0.431 | 146.022 |
| | | ReSample | 29.950 | 0.842 | 0.201 | 278.498 | 26.916 | 0.756 | 0.255 | 315.707 |
| | | Ours | **30.091** | **0.877** | **0.201** | 45.335 | **27.245** | **0.775** | 0.288 | 46.454 |
| Gaussian deblurring | Pixel | DAPS | 28.895 | 0.775 | 0.253 | 50.400 | 25.946 | 0.662 | 0.352 | 94.605 |
| | | SITCOM | 28.775 | 0.820 | 0.261 | 32.841 | 26.201 | 0.702 | 0.351 | 103.338 |
| | | C-IIGDM | 24.432 | 0.678 | 0.368 | 1.305 | 23.701 | 0.595 | 0.352 | 4.881 |
| | | HRDIS | 27.674 | 0.791 | 0.259 | 2.569 | 24.575 | 0.633 | 0.419 | 5.960 |
| | | Ours | **29.406** | **0.836** | **0.247** | 2.612 | 26.181 | **0.705** | **0.344** | 4.958 |
| | Latent | LatentDAPS | 25.742 | 0.732 | 0.384 | 93.313 | 22.818 | 0.561 | 0.543 | 98.340 |
| | | PSLD | 16.807 | 0.227 | 0.569 | 94.823 | 16.608 | 0.212 | 0.566 | 148.738 |
| | | ReSample | 26.345 | 0.661 | 0.329 | 292.612 | 23.530 | 0.497 | 0.439 | 333.822 |
| | | Ours | **28.006** | **0.793** | **0.312** | 46.307 | **25.356** | **0.661** | **0.424** | 48.229 |
| Motion deblurring | Pixel | DAPS | 31.074 | 0.829 | 0.199 | 50.924 | 28.838 | 0.776 | 0.243 | 94.681 |
| | | SITCOM | 31.172 | 0.872 | 0.203 | 36.684 | 28.875 | 0.807 | 0.247 | 103.338 |
| | | Ours | **31.736** | **0.878** | **0.171** | 2.616 | **29.037** | 0.799 | **0.236** | 4.623 |
| | Latent | LatentDAPS | 26.649 | 0.757 | 0.361 | 93.400 | 23.557 | 0.592 | 0.513 | 97.988 |
| | | PSLD | 19.237 | 0.288 | 0.518 | 90.682 | 18.327 | 0.288 | 0.544 | 148.151 |
| | | ReSample | 28.744 | 0.754 | 0.262 | 302.828 | 24.845 | 0.579 | 0.404 | 316.985 |
| | | Ours | **29.285** | **0.822** | 0.278 | 46.785 | **26.627** | **0.709** | **0.386** | 47.282 |
| Phase retrieval | Pixel | DAPS | **30.253** | 0.801 | 0.202 | 122.100 | **22.354** | 0.519 | 0.402 | 320.926 |
| | | SITCOM | 28.512 | 0.791 | 0.240 | 37.425 | 18.704 | 0.393 | 0.519 | 99.103 |
| | | HRDIS | 23.670 | 0.537 | 0.448 | 12.020 | 14.019 | 0.195 | 0.722 | 29.915 |
| | | Ours | 29.253 | **0.851** | 0.218 | 10.354 | 19.738 | 0.490 | 0.479 | 16.629 |
| | Latent | LatentDAPS | 23.450 | 0.695 | 0.418 | 193.005 | 17.067 | 0.446 | 0.624 | 202.426 |
| | | ReSample | 24.676 | 0.606 | 0.412 | 321.227 | 16.913 | 0.320 | 0.608 | 354.430 |
| | | Ours | **28.330** | **0.789** | **0.244** | 87.167 | **18.874** | 0.441 | 0.507 | 85.520 |
| Nonlinear deblur | Pixel | DAPS | 28.907 | 0.780 | 0.222 | 763.863 | 27.537 | 0.734 | 0.266 | 1453.314 |
| | | SITCOM | **29.770** | **0.844** | **0.190** | 43.040 | **28.138** | **0.791** | **0.218** | 113.165 |
| | | HRDIS | 24.929 | 0.658 | 0.357 | 3.094 | 22.553 | 0.504 | 0.448 | 6.653 |
| | | Ours | 27.818 | 0.803 | 0.280 | 57.903 | 25.607 | 0.695 | 0.373 | 62.350 |
| | Latent | LatentDAPS | 25.151 | 0.727 | 0.384 | 229.700 | 22.516 | 0.568 | 0.530 | 249.639 |
| | | ReSample | 28.748 | 0.797 | **0.236** | 1276.326 | 26.047 | 0.697 | **0.301** | 1250.783 |
| | | Ours | 28.746 | **0.823** | 0.260 | 110.567 | 26.234 | 0.720 | 0.350 | 113.537 |
| High dynamic range | Pixel | DAPS | 26.988 | 0.834 | 0.196 | 103.243 | 26.568 | 0.819 | **0.198** | 293.286 |
| | | SITCOM | **27.628** | 0.808 | 0.214 | 38.150 | **26.849** | 0.796 | 0.207 | 109.946 |
| | | HRDIS | 26.346 | 0.836 | **0.178** | 2.428 | 24.623 | **0.825** | 0.199 | 5.989 |
| | | Ours | 26.275 | **0.843** | 0.218 | 7.212 | 24.522 | 0.775 | 0.290 | 13.367 |
| | Latent | LatentDAPS | 20.789 | 0.630 | 0.512 | 197.250 | 19.394 | 0.469 | 0.641 | 207.469 |
| | | ReSample | 25.038 | 0.822 | **0.239** | 261.558 | **24.950** | **0.783** | **0.257** | 285.495 |
| | | Ours | **25.869** | **0.832** | 0.247 | 83.790 | 24.415 | 0.773 | 0.291 | 85.685 |

**Implementation details.** For all experiments, we employ pretrained diffusion models in both pixel and latent domains. For the pixel-space setting, we use diffusion models trained on FFHQ (Chung et al. (2023a)) and ImageNet (Dhariwal & Nichol (2021)). For the latent-space setting, we use the unconditional LDM-VQ4 model (Rombach et al. (2022)) for both FFHQ and ImageNet. These models are used consistently across all baselines and our method to ensure a fair comparison. We adopt the time-step discretization and noise schedule from EDM (Karras et al. (2022)). Evaluation is performed on 100 images from FFHQ ($256 \times 256$) and 100 images from ImageNet ($256 \times 256$).

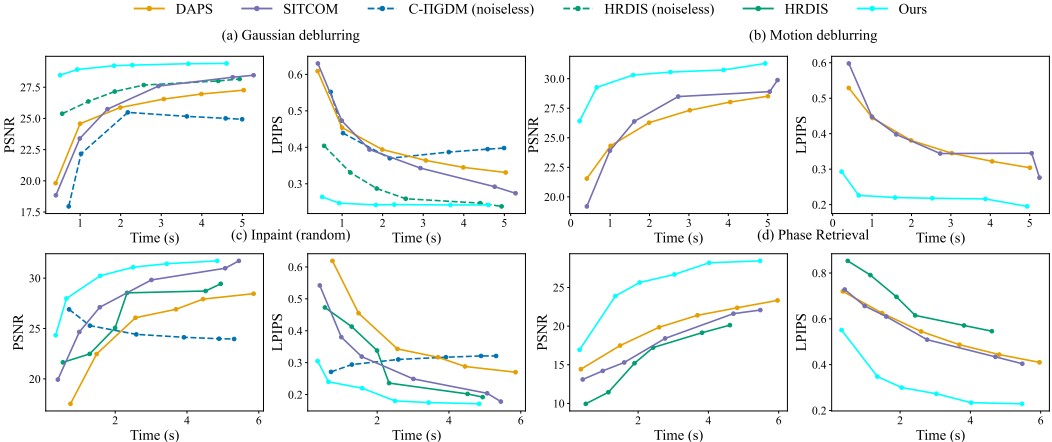

Figure 3: Quantitative evaluations comparing image quality and computational time for baseline methods. Each point is derived from an experiment on 100 FFHQ images. The y-axis value (PSNR or LPIPS) is the mean of the scores from the 100 resulting images. The x-axis value is the average per-image run-time, calculated by dividing the total processing time for all 100 images by 100. The plots show results for three linear tasks (a-c) and one nonlinear task (d).

Across all tasks, measurements are corrupted by additive Gaussian noise with a standard deviation of $\beta = 0.05$, and performance is reported using PSNR, SSIM, and LPIPS. Inputs are in the range $[-1, 1]$ for PSNR/LPIPS and $[0, 1]$ for SSIM. For LPIPS, we keep the default max-pooling layers in the VGG-based implementation. Additional experimental details are provided in Appendix A.6.

**Baselines.** We compare our method against a range of state-of-the-art baselines in both pixel and latent spaces. In pixel space, we include recent fast-sampling methods such as SITCOM (Alkhouri et al. (2025)), C-ΠGDM (Pandey et al. (2024)), and HRDIS (Dou et al. (2025)), alongside DAPS (Zhang et al. (2025)), which is recognized for its balance of performance and efficiency. For latent-space comparisons, we benchmark against prominent methods including PSLD (Rout et al. (2023)), ReSample (Song et al. (2024)), and Latent-DAPS (Zhang et al. (2025)). Details of the baseline methods are provided in Appendix A.7.

## 3.2 MAIN RESULTS

Table 1 presents the quantitative results on the FFHQ and ImageNet datasets, where all baselines are run with their official default settings. In pixel space, our method achieves comparable or superior performance to the baselines across nearly all tasks on both datasets, but with a significantly lower run-time. This acceleration is particularly evident in Gaussian and motion deblurring, where FAST-DIPS is about $19.4\times$ faster than DAPS on FFHQ and about $20.8\times$ faster than SITCOM on ImageNet, while achieving comparable PSNR/SSIM/LPIPS. For the challenging nonlinear task of phase retrieval, we follow the common practice of selecting the best of four independent runs. In this setting, our method is approximately $11.8\times$ faster than DAPS on FFHQ and $19.3\times$ faster on ImageNet, achieving higher SSIM on FFHQ and the second-best SSIM/LPIPS among pixel-space baselines on ImageNet. Furthermore, our approach addresses key inefficiencies commonly found in latent-space methods. While most guided techniques suffer from long run-times due to the computational cost of backpropagating through the decoder, our hybrid pixel–latent schedule avoids this bottleneck. By performing corrections in pixel space during the early sampling stages and switching to latent-space correction later, our method effectively reduces sampling time while maintaining high-quality, manifold-faithful reconstructions across datasets.

Table 1 alone does not fully capture how different methods compare under the same run-time budget. To offer a more comprehensive evaluation, Figure 3 reports PSNR and LPIPS while considering the computational run-time. For this benchmark, we vary only the number of sampling steps/inner iterations per method, while all other hyperparameters were kept at their originally proposed optimal values to ensure a fair comparison. (Full details are provided in Appendix A.7). We evaluate

three linear and one nonlinear task in total. Across all four tasks, our method improves steadily as run-time increases while maintaining a clear gap over competing baselines. The advantage is particularly pronounced in motion deblurring and phase retrieval, where the superiority highlighted earlier is equally evident under identical run-time budgets. In Gaussian deblurring, even compared to noiseless baselines, our method quickly attains strong PSNR and LPIPS in the early stage and sustains or further improves them as sampling proceeds. The same trend is also observed in random inpainting. For this task, perceptual quality is paramount, and our method generates natural-looking results while consistently maintaining low LPIPS.

We include additional experiments on FAST-DIPS in Appendix A.8, covering the effectiveness of the $\mathbf{x}$-update step, hyperparameter robustness, the hybrid schedule trade-off, experiments with non-Gaussian noise and qualitative results in both pixel and latent spaces.

### 3.3 ABLATION STUDIES

We study two factors inside the per-level correction: whether we enforce feasibility by projection and how we choose the step size for the $\mathbf{x}$-update. The projection variant is our default FAST-DIPS (ADMM + proj.); the no-projection control is an unsplit penalized solver we call QDP (no splitting, no proj.), which minimizes the same quadratic objective as the ADMM $\mathbf{x}$-subproblem. To compare fairly, we equalize compute by counting first-order autodiff work: each $\mathbf{x}$-gradient step uses one forward of $A$, one VJP, and one JVP (or a single forward probe for FD); projection and dual updates are negligible. With $K$ ADMM iterations and $S$ gradient steps per iteration, FAST-DIPS spends $K \times S$ such triplets at each diffusion level, so we give QDP exactly $K \times S$ gradient steps per level. For step size we compare a tuned constant $\alpha$, the analytic model-optimal $\alpha^\star$ (one VJP + one JVP), and a forward-only finite-difference surrogate $\alpha_{\mathrm{FD}}$. Full protocol and numbers are provided in Appendix A.4, Table 2.

On a representative linear pixel task (Gaussian blur), $\alpha_{\mathrm{FD}}$ reaches virtually the same quality as $\alpha^\star$ at lower cost; on the nonlinear latent HDR task the optimization is sensitive to a fixed step and the JVP-based $\alpha^\star$ is the robust choice, whereas $\alpha_{\mathrm{FD}}$ tends to underperform through the decoder–forward stack. Enforcing feasibility by projection consistently improves quality relative to the unsplit penalty path under the matched budget; the extra cost in latent space is dominated by backprop through the decoder rather than the projection. A practical recipe is therefore to use $\alpha_{\mathrm{FD}}$ in pixel space and $\alpha^\star$ in latent space within FAST-DIPS.

## 4 CONCLUSION

Our proposed method, FAST-DIPS, targets practical challenges in training-free, diffusion-based inverse problems. It is broadly applicable: by using VJP and JVP from automatic differentiation instead of a hand-crafted adjoint, it can handle a wide range of linear and nonlinear forward models without requiring SVDs or pseudo-inverses.

For the guidance step, we replace generic optimizers (e.g., Adam with tuned learning rates) by a single gradient update with an analytic step size from a local quadratic model. This deterministic update removes step-size hyperparameters and improves efficiency and stability.

Empirically, FAST-DIPS shows a predictable trade-off between computation and quality: more correction steps consistently improve reconstructions. The framework also does not rely on carefully chosen initial samples. Limitations and future directions are discussed in Appendix A.9.

## 5 REPRODUCIBILITY STATEMENT

Our experimental setup (datasets, pretrained models, forward operators, noise levels, metrics, and hardware) is specified in Section 3.1. In brief, we use publicly available pixel- and latent-space diffusion priors on the FFHQ-256 and the ImageNet-256, the EDM discretization, additive Gaussian measurement noise with $\beta = 0.05$, and evaluate PSNR/SSIM/LPIPS on 100 images. Most experiments are performed using a single NVIDIA RTX 4090 GPU, whereas the experiments reported in Tables 5, 7, 9 were conducted on a single RTX 6000 Ada GPU. For phase retrieval, we follow the common "best-of-4" protocol for all methods (ours and baselines). Baselines are run from the authors' official repositories with their recommended defaults; Appendix A.7 lists the packages we used and task-specific settings. We set the random seed to 42 for all experiments, including all baselines. For evaluation, we compute PSNR with skimage.metrics (Van der Walt et al. (2014)), SSIM with TorchMetrics (Detlefsen et al. (2022)), and LPIPS with the lpips library (VGG backbone) (Zhang et al. (2018)).

## ACKNOWLEDGEMENT

This work was supported in part by the National Research Foundation of Korea (NRF) grant funded by the Korea government (MSIT) (RS-2025-24683103, RS-2026-25476632), in part by Korea Basic Science Institute (National research Facilities and Equipment Center) grant funded by the Ministry of Science and ICT (No. RS-2024-00401899), and in part by Institute of Information & communications Technology Planning & Evaluation (IITP) under the Leading Generative AI Human Resources Development (IITP-2026-RS-2024-00360227) grant funded by the Korea government (MSIT).

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

# A APPENDIX

## A.1 LATENT-SPACE FAST-DIPS AND A HYBRID PIXEL–LATENT SCHEDULE

The pixel-space method in Section 2.5 corrects the denoiser's proposal directly in image space. In many diffusion systems, however, the prior is trained in a lower-dimensional latent space. Let $\mathcal{E} : \mathbb{R}^{CHW} \to \mathbb{R}^k$ and $\mathcal{D} : \mathbb{R}^k \to \mathbb{R}^{CHW}$ denote a pretrained encoder–decoder with $\mathbf{z}_0 = \mathcal{E}(\mathbf{x}_0)$ and $\mathbf{x}_0 = \mathcal{D}(\mathbf{z}_0)$. Measurements are still acquired in pixel space via Equation 1. A latent denoiser $\mathbf{z}_{\mathrm{den}}(\mathbf{z}_t, \sigma_t)$ is available from the diffusion prior. We now derive a latent analogue of the per-level objective and show that the pixel-space construction transfers under the substitution $\mathcal{A} \mapsto \mathcal{A} \circ \mathcal{D}$ and $\mathbf{x} \leftrightarrow \mathbf{z}$.

**Per-level surrogate in latent space.** At level $t$, the denoiser proposes $\mathbf{z}_{0|t} := \mathbf{z}_{\mathrm{den}}(\mathbf{z}_t, \sigma_t)$. As in Section 2.3, we approximate $p(\mathbf{z}_0 \mid \mathbf{z}_t)$ by a local Gaussian centered at $\mathbf{z}_{0|t}$ with variance parameter $\gamma_z > 0$ (we use $\gamma_z = \sigma_t^2$ for schedule-awareness), and we employ the same set-valued / hard-constraint likelihood surrogate in the measurement space, now expressed through the decoder:

$$\tilde{p}_t(\mathbf{z}_0 \mid \mathbf{z}_t, \mathbf{y}) \propto \exp\left(-\frac{1}{2\gamma_z}\|\mathbf{z}_0 - \mathbf{z}_{0|t}\|^2\right) \mathbf{1}\{\|\mathcal{A}(\mathcal{D}(\mathbf{z}_0)) - \mathbf{y}\| \leq \varepsilon_z\}. \tag{23}$$

Since feasibility is enforced in the same measurement space, in our experiments we simply set $\varepsilon_z = \varepsilon$ (unless noted otherwise). Taking the mode yields the latent per-level MAP:

$$\mathbf{z}_{0|t}^{\mathrm{corr}} \in \arg\min_{\mathbf{z} \in \mathbb{R}^k} \frac{1}{2\gamma_z}\|\mathbf{z} - \mathbf{z}_{0|t}\|^2 \ \text{ s.t. } \ \|\mathcal{A}(\mathcal{D}(\mathbf{z})) - \mathbf{y}\| \leq \varepsilon_z. \tag{24}$$

Re-annealing then follows the same transport rule as Equation 11:

$$\mathbf{z}_{t-1} = \mathbf{z}_{0|t}^{\mathrm{corr}} + \sigma_{t-1}\boldsymbol{\xi}, \qquad \boldsymbol{\xi} \sim \mathcal{N}(\mathbf{0}, I), \qquad \mathbf{x}_{t-1} = \mathcal{D}(\mathbf{z}_{t-1}). \tag{25}$$

**ADMM in latent space and adjoint-free updates.** Introduce $\mathbf{v} \approx \mathcal{A}(\mathcal{D}(\mathbf{z}))$ and the same feasibility set $\mathcal{C} := \{\mathbf{v} : \|\mathbf{v} - \mathbf{y}\| \leq \varepsilon_z\}$. The scaled ADMM iterations mirror Equation 13–Equation 15:

$$\mathbf{z}^{k+1} = \arg\min_{\mathbf{z}} \frac{1}{2\gamma_z}\|\mathbf{z} - \mathbf{z}_{0|t}\|^2 + \frac{\rho_z}{2}\|\mathcal{A}(\mathcal{D}(\mathbf{z})) - \mathbf{v}^k + \mathbf{u}^k\|^2, \tag{26}$$

$$\mathbf{v}^{k+1} = \Pi_{\mathcal{C}}\left(\mathcal{A}(\mathcal{D}(\mathbf{z}^{k+1})) + \mathbf{u}^k\right), \tag{27}$$

$$\mathbf{u}^{k+1} = \mathbf{u}^k + \mathcal{A}(\mathcal{D}(\mathbf{z}^{k+1})) - \mathbf{v}^{k+1}. \tag{28}$$

The projection $\Pi_{\mathcal{C}}$ is identical to Equation 16 because feasibility is enforced in measurement space. As in pixel space, in practice we run only a small fixed number of iterations and do not solve the $\mathbf{z}$-subproblem Equation 26 exactly; instead we use one (or a few) adjoint-free gradient steps on $F_z$ with analytic step initialization and backtracking. For the $\mathbf{z}$-update, define

$$F_z(\mathbf{z}) = \frac{1}{2\gamma_z}\|\mathbf{z} - \mathbf{z}_{0|t}\|^2 + \frac{\rho_z}{2}\|\mathcal{A}(\mathcal{D}(\mathbf{z})) - \mathbf{b}^k\|^2, \qquad \mathbf{b}^k := \mathbf{v}^k - \mathbf{u}^k, \tag{29}$$

and let $\mathbf{r} := \mathcal{A}(\mathcal{D}(\mathbf{z})) - \mathbf{b}^k$. Its gradient is

$$\mathbf{g}_z = \nabla F_z(\mathbf{z}) = \frac{1}{\gamma_z}(\mathbf{z} - \mathbf{z}_{0|t}) + \rho_z J_{\mathcal{A}\circ\mathcal{D}}(\mathbf{z})^\top\left(\mathcal{A}(\mathcal{D}(\mathbf{z})) - \mathbf{b}^k\right), \qquad \mathbf{z} \leftarrow \mathbf{z} - \alpha\,\mathbf{g}_z. \tag{30}$$

As in pixel space, both the VJP $J_{\mathcal{A}\circ\mathcal{D}}(\mathbf{z})^\top\mathbf{r}$ and the JVP $J_{\mathcal{A}\circ\mathcal{D}}(\mathbf{z})\mathbf{g}_z$ are obtained directly from autodiff (backprop through $\mathcal{D}$ and $\mathcal{A}$; forward-mode or a single finite-difference for the JVP if needed), so the update remains adjoint-free.

**Analytic step size in latent space.** Let $\mathbf{s}_z := \mathbf{z} - \mathbf{z}_{0|t}$. Linearizing $\mathcal{A} \circ \mathcal{D}$ along $-\mathbf{g}_z$ gives $\mathcal{A}(\mathcal{D}(\mathbf{z} - \alpha\mathbf{g}_z)) \approx \mathcal{A}(\mathcal{D}(\mathbf{z})) - \alpha\,J_{\mathcal{A}\circ\mathcal{D}}(\mathbf{z})\,\mathbf{g}_z$. The scalar quadratic model

$$\tilde{F}_z(\alpha) = \frac{1}{2\gamma_z}\|\mathbf{s}_z - \alpha\mathbf{g}_z\|^2 + \frac{\rho_z}{2}\|\mathbf{r} - \alpha J_{\mathcal{A}\circ\mathcal{D}}(\mathbf{z})\mathbf{g}_z\|^2 \tag{31}$$

is minimized at

$$\alpha_z^* = \frac{\frac{1}{\gamma_z}\langle \mathbf{s}_z, \mathbf{g}_z\rangle + \rho_z\langle \mathbf{r}, J_{\mathcal{A}\circ\mathcal{D}}(\mathbf{z})\mathbf{g}_z\rangle}{\frac{1}{\gamma_z}\|\mathbf{g}_z\|^2 + \rho_z\|J_{\mathcal{A}\circ\mathcal{D}}(\mathbf{z})\mathbf{g}_z\|^2} \tag{32}$$

followed by backtracking to ensure descent of $F_z$; if $\mathbf{g}_z = \mathbf{0}$ we are already stationary for the current $\mathbf{z}$-subproblem and set $\alpha = 0$.

**Proposition 2** (Latent model-optimal step and descent). *Under $C^1$ regularity of $\mathcal{A} \circ \mathcal{D}$ near $\mathbf{z}$ and local Lipschitzness of its Jacobian, if $\mathbf{g}_z \neq \mathbf{0}$ then the step $\alpha_z^*$ in Equation 32 minimizes the quadratic model $\tilde{F}_z(\alpha)$. Moreover, using $\alpha_z^*$ (or its positive clamp) to initialize an Armijo backtracking line search guarantees that the accepted step yields monotone decrease of the current latent $\mathbf{z}$-subproblem objective $F_z$ in Equation 29 (with $\mathbf{b}^k$ fixed). If $\mathbf{g}_z = \mathbf{0}$, then $\mathbf{z}$ is stationary for $F_z$ and no step is taken.*

**Proposition 3** (KKT at latent ADMM fixed points). *If $(\mathbf{z}^*, \mathbf{v}^*, \mathbf{u}^*)$ is a fixed point of the exact scaled-ADMM updates Equation 26–Equation 28 (i.e., $\mathbf{z}^{k+1}$ solves the $\mathbf{z}$-subproblem Equation 26 exactly), then $\mathbf{z}^*$ satisfies the KKT conditions of the latent proximal problem Equation 24.*

**Remark 2** (Transfer of pixel-space results). *All pixel-space results in §2.4–§2.6 transfer to the latent case by replacing $\mathcal{A}$ with $\mathcal{A} \circ \mathcal{D}$ and $\mathbf{x}$ with $\mathbf{z}$. The projection statement is unchanged because feasibility is enforced in measurement space. Propositions 2 and 3 follow from the corresponding pixel-space arguments by the chain rule. Proposition 6 also applies to the latent Gaussian injected distributions in Equation 25 by replacing $\mathbf{x}$ with $\mathbf{z}$; after deterministic decoding, KL cannot increase (data processing).*

**Why (and when) prefer latent updates.** Late in the schedule, $\sigma_t$ is small, the denoiser's latent prediction $\mathbf{z}_{0|t}$ lies near the generative manifold, and optimizing in $\mathbf{z}$ respects that geometry by construction. Early in the schedule, however, correcting in pixel space is often cheaper (no backprop through $\mathcal{D}$) and sufficiently robust because injected noise dominates the time–marginal. This observation motivates a hybrid schedule.

**Hybrid pixel–latent schedule.** We adopt a single switching parameter $\sigma_{\mathrm{switch}}$: for $\sigma_t > \sigma_{\mathrm{switch}}$ we correct in pixel space using Equation 9–Equation 15, then re-encode $\mathbf{z} \leftarrow \mathcal{E}(\mathbf{x})$ before annealing in latent space; once $\sigma_t \leq \sigma_{\mathrm{switch}}$, we correct directly in latent space using Equation 24–Equation 28. This keeps early iterations light and late iterations manifold-faithful.

**Complexity and switching.** A latent $\mathbf{z}$-gradient step costs one pass through $\mathcal{D}$ and $\mathcal{A}$ to form $\mathbf{r}$, one VJP through $\mathcal{A} \circ \mathcal{D}$ to form $J_{\mathcal{A} \circ \mathcal{D}}^\top \mathbf{r}$, and one JVP to form $J_{\mathcal{A} \circ \mathcal{D}} \mathbf{g}_z$; we found this JVP-based step is effective for nonlinear-deblur in latent space. In pixel space, for strongly nonlinear $\mathcal{A}$ we recommend the FD variant Equation 22 and 38, which swaps the JVP for a single extra forward call and was both faster and more stable in our nonlinear-deblur experiments. The switch $\sigma_{\mathrm{switch}}$ trades early-time efficiency for late-time fidelity; a stable default is to place it where the SNR of the denoiser's prediction visibly improves (e.g., where $\gamma_t$ becomes comparable to the scale of $\|\mathbf{x} - \mathbf{x}_{0|t}\|$ in Equation 17).

**Remark 3** (Consistency of pixel $\to$ encode with latent correction). *If $\mathcal{E}$ and $\mathcal{D}$ are approximately inverses near the data manifold (i.e., $\mathcal{D}(\mathcal{E}(\mathbf{x})) \approx \mathbf{x}$ and $\mathcal{E}(\mathcal{D}(\mathbf{z})) \approx \mathbf{z}$) and are locally Lipschitz, then a pixel correction followed by $\mathbf{z} \leftarrow \mathcal{E}(\mathbf{x})$ produces a latent iterate whose discrepancy is controlled by the local reconstruction error of the autoencoder (i.e., how close $\mathcal{D} \circ \mathcal{E}$ is to $\mathrm{Id}$ near the manifold). Thus the hybrid scheme is a coherent approximation of the pure latent method early in the schedule.*

## A.2 ALGORITHMS

---

**Algorithm 1** FAST-DIPS in Pixel Space

---

**Require:** measurement $\mathbf{y}$; schedule $\{\sigma_t\}$; denoiser $\mathbf{x}_{\text{den}}(\cdot, \sigma_t)$; forward $\mathcal{A}$; parameters $\rho, \{\gamma_t\}, K,$
    $S, \eta, \epsilon$
**Ensure:** reconstructed image $\mathbf{x}_0$
1: Sample $\mathbf{x}_T \sim \mathcal{N}(\mathbf{0}, \sigma_T^2 I)$
2: **for** $t = T$ **down to** $1$ **do**
3:    *predict* $\mathbf{x}_{0|t} \leftarrow \mathbf{x}_{\text{den}}(\mathbf{x}_t, \sigma_t)$
4:    Initialize $\mathbf{x} \leftarrow \mathbf{x}_{0|t};$   $\mathbf{v} \leftarrow \mathcal{A}(\mathbf{x});$   $\mathbf{u} \leftarrow \mathbf{0}$
5:    **for** $k = 1$ **to** $K$ **do**
6:       $\mathbf{b} \leftarrow \mathbf{v} - \mathbf{u};$   $F(\mathbf{x}) \leftarrow \frac{1}{2\gamma_t}\|\mathbf{x} - \mathbf{x}_{0|t}\|^2 + \frac{\rho}{2}\|\mathcal{A}(\mathbf{x}) - \mathbf{b}\|^2$
7:       **for** $i = 1$ **to** $S$ **do**                  $\triangleright$ **x**-update: gradient step + backtracking
8:          $\mathbf{r} \leftarrow \mathcal{A}(\mathbf{x}) - \mathbf{b}; \mathbf{s} \leftarrow \mathbf{x} - \mathbf{x}_{0|t}$
9:          $\mathbf{g}_{\text{data}} \leftarrow \nabla_{\mathbf{x}}\left(\frac{1}{2}\|\mathcal{A}(\mathbf{x}) - \mathbf{b}\|^2\right)$          $\triangleright$ via automatic differentiation
10:        $\mathbf{g} \leftarrow \frac{1}{\gamma_t}\mathbf{s} + \rho\,\mathbf{g}_{\text{data}}; \Delta\mathcal{A} \leftarrow \mathcal{A}(\mathbf{x} + \eta\mathbf{g}) - \mathcal{A}(\mathbf{x})$
11:        $\alpha \leftarrow \dfrac{\eta^2\frac{1}{\gamma_t}\langle\mathbf{s}, \mathbf{g}\rangle + \eta\,\rho\,\langle\mathbf{r}, \Delta\mathcal{A}\rangle}{\eta^2\frac{1}{\gamma_t}\|\mathbf{g}\|^2 + \rho\,\|\Delta\mathcal{A}\|^2}$
12:        Backtrack on $\alpha$ until $F(\mathbf{x} - \alpha\mathbf{g}) < F(\mathbf{x})$; set $\mathbf{x} \leftarrow \mathbf{x} - \alpha\mathbf{g}$
13:       **end for**
14:       $\mathbf{w} \leftarrow \mathcal{A}(\mathbf{x}) + \mathbf{u};$   $\mathbf{v} \leftarrow \Pi_{\|\cdot - \mathbf{y}\| \leq \epsilon}(\mathbf{w})$
15:       $\mathbf{u} \leftarrow \mathbf{u} + \mathcal{A}(\mathbf{x}) - \mathbf{v}$
16:    **end for**
17:    Sample $\boldsymbol{\xi} \sim \mathcal{N}(\mathbf{0}, I)$ and set $\mathbf{x}_{t-1} \leftarrow \mathbf{x} + \sigma_{t-1}\boldsymbol{\xi}$
18: **end for**
19: **return** $\mathbf{x}_0$

---

---

**Algorithm 2** FAST-DIPS in Latent Space

---

**Require:** measurement $\mathbf{y}$; schedule $\{\sigma_t\}$; latent denoiser $\mathbf{z}_{\mathrm{den}}(\cdot, \sigma_t)$; encoder $\mathcal{E}$; decoder $\mathcal{D}$; forward $\mathcal{A}$; parameters $\rho_x, \gamma_x, K_x, S_x, \varepsilon_x,\ \rho_z, \gamma_z, K_z, S_z, \varepsilon_z,\ \sigma_{\mathrm{switch}}$
**Ensure:** reconstructed image $\mathbf{x}_0$
 1: Sample $\mathbf{z}_T \sim \mathcal{N}(\mathbf{0}, \sigma_T^2 I)$
 2: **for** $t = T$ **down to** $1$ **do**
 3:     *predict (latent)* $\mathbf{z}_{0|t} \leftarrow \mathbf{z}_{\mathrm{den}}(\mathbf{z}_t, \sigma_t)$
 4:     **if** $\sigma_t > \sigma_{\mathrm{switch}}$ **then**                             $\triangleright$ early: pixel correction
 5:         $\mathbf{x}_{0|t} \leftarrow \mathcal{D}(\mathbf{z}_{0|t});\ \ \mathbf{x} \leftarrow \mathbf{x}_{0|t};\ \ \mathbf{v} \leftarrow \mathcal{A}(\mathbf{x});\ \ \mathbf{u} \leftarrow \mathbf{0}$
 6:         **for** $k = 1$ **to** $K_x$ **do**
 7:             $\mathbf{b} \leftarrow \mathbf{v} - \mathbf{u};\ \ \ F_x(\mathbf{x}) \leftarrow \frac{1}{2\gamma_x}\|\mathbf{x} - \mathbf{x}_{0|t}\|^2 + \frac{\rho_x}{2}\|\mathcal{A}(\mathbf{x}) - \mathbf{b}\|^2$
 8:             **for** $s = 1$ **to** $S_x$ **do**                 $\triangleright$ x-update with analytic step
 9:                 $\mathbf{g} \leftarrow \frac{1}{\gamma_x}(\mathbf{x} - \mathbf{x}_{0|t}) + \rho_x J_{\mathcal{A}}(\mathbf{x})^\top\big(\mathcal{A}(\mathbf{x}) - \mathbf{b}\big)$
10:                 Form $J_{\mathcal{A}}(\mathbf{x})\mathbf{g}$ (JVP) and set $\alpha$ by Equation 21;
11:                 Backtrack on $\alpha$ until $F_x(\mathbf{x} - \alpha\mathbf{g}) < F_x(\mathbf{x})$; set $\mathbf{x} \leftarrow \mathbf{x} - \alpha\mathbf{g}$
12:             **end for**
13:             $\mathbf{w} \leftarrow \mathcal{A}(\mathbf{x}) + \mathbf{u};\ \ \ \mathbf{v} \leftarrow \Pi_{\|\cdot - \mathbf{y}\| \leq \varepsilon_x}(\mathbf{w});\ \ \ \mathbf{u} \leftarrow \mathbf{u} + \mathcal{A}(\mathbf{x}) - \mathbf{v}$
14:         **end for**
15:         *re-encode* $\mathbf{z} \leftarrow \mathcal{E}(\mathbf{x})$
16:     **else**                                       $\triangleright$ late: latent correction
17:         $\mathbf{z} \leftarrow \mathbf{z}_{0|t};\ \ \mathbf{v} \leftarrow \mathcal{A}(\mathcal{D}(\mathbf{z}));\ \ \mathbf{u} \leftarrow \mathbf{0}$
18:         **for** $k = 1$ **to** $K_z$ **do**
19:             $\mathbf{b} \leftarrow \mathbf{v} - \mathbf{u};\ \ \ F_z(\mathbf{z}) \leftarrow \frac{1}{2\gamma_z}\|\mathbf{z} - \mathbf{z}_{0|t}\|^2 + \frac{\rho_z}{2}\|\mathcal{A}(\mathcal{D}(\mathbf{z})) - \mathbf{b}\|^2$
20:             **for** $s = 1$ **to** $S_z$ **do**                 $\triangleright$ z-update with analytic step
21:                 $\mathbf{g}_z \leftarrow \frac{1}{\gamma_z}(\mathbf{z} - \mathbf{z}_{0|t}) + \rho_z J_{\mathcal{A}\circ\mathcal{D}}(\mathbf{z})^\top\big(\mathcal{A}(\mathcal{D}(\mathbf{z})) - \mathbf{b}\big)$
22:                 Form $J_{\mathcal{A}\circ\mathcal{D}}(\mathbf{z})\mathbf{g}_z$ (JVP) and set $\alpha$ by Equation 32;
23:                 Backtrack on $\alpha$ until $F_z(\mathbf{z} - \alpha\mathbf{g}_z) < F_z(\mathbf{z})$; set $\mathbf{z} \leftarrow \mathbf{z} - \alpha\mathbf{g}_z$
24:             **end for**
25:             $\mathbf{w} \leftarrow \mathcal{A}(\mathcal{D}(\mathbf{z})) + \mathbf{u};\ \ \ \mathbf{v} \leftarrow \Pi_{\|\cdot - \mathbf{y}\| \leq \varepsilon_z}(\mathbf{w});\ \ \ \mathbf{u} \leftarrow \mathbf{u} + \mathcal{A}(\mathcal{D}(\mathbf{z})) - \mathbf{v}$
26:         **end for**
27:     **end if**
28:     *re-anneal* $\mathbf{z}_{t-1} \leftarrow \mathbf{z} + \sigma_{t-1}\boldsymbol{\xi},\ \boldsymbol{\xi} \sim \mathcal{N}(\mathbf{0}, I)$
29: **end for**
30: **return** $\mathbf{x}_0 \leftarrow \mathcal{D}(\mathbf{z}_0)$

---

## A.3 THEORY AND PROOFS

This appendix first summarizes the proposed FAST-DIPS procedure and its modeling assumptions (App. A.3.1). We then restate and prove the pixel-space results used by the method (App. A.3.2). For the latent variant, the corresponding statements follow from the same arguments by the substitution $\mathcal{A} \mapsto \mathcal{A} \circ \mathcal{D}$ and the chain rule; we provide a brief transfer explanation and omit redundant proofs (App. A.3.3). Finally, we give step-by-step derivations of the analytic step sizes used in the pixel and latent updates and explain how they can be computed with autodiff VJP/JVP or a single forward-difference probe (App. A.3.4).

### A.3.1 OVERVIEW AND ASSUMPTIONS

**Method in one paragraph.** At diffusion level $t$, the pretrained denoiser returns an anchor $\mathbf{x}_{0|t} = \mathbf{x}_{\mathrm{den}}(\mathbf{x}_t, \sigma_t)$. We then solve a hard-constrained proximal problem around $\mathbf{x}_{0|t}$,

$$\min_{\mathbf{x} \in \mathbb{R}^{CHW}}\ \frac{1}{2\gamma_t}\|\mathbf{x} - \mathbf{x}_{0|t}\|^2 \quad \text{s.t.} \quad \|\mathcal{A}(\mathbf{x}) - \mathbf{y}\| \leq \varepsilon, \tag{33}$$

in the standard (Euclidean) measurement space. We solve Equation 33 by a scaled ADMM-style augmented Lagrangian splitting with variables $(\mathbf{x}, \mathbf{v}, \mathbf{u})$:

$$\mathbf{x}^{k+1} = \arg\min_{\mathbf{x}} \frac{1}{2\gamma_t} \|\mathbf{x} - \mathbf{x}_{0|t}\|^2 + \frac{\rho}{2} \|\mathcal{A}(\mathbf{x}) - \mathbf{v}^k + \mathbf{u}^k\|^2, \tag{34}$$

$$\mathbf{v}^{k+1} = \Pi_{\mathcal{C}}\big(\mathcal{A}(\mathbf{x}^{k+1}) + \mathbf{u}^k\big), \quad \mathcal{C} = \{\mathbf{v} : \|\mathbf{v} - \mathbf{y}\| \leq \varepsilon\}, \tag{35}$$

$$\mathbf{u}^{k+1} = \mathbf{u}^k + \mathcal{A}(\mathbf{x}^{k+1}) - \mathbf{v}^{k+1}. \tag{36}$$

The $\mathbf{v}$-update is a closed-form projection onto a ball; the $\mathbf{x}$-update is one (or a few) adjoint-free gradient steps with an analytic, model-optimal step size, where the needed directional Jacobian term $J_{\mathcal{A}}(\mathbf{x})\mathbf{g}$ is obtained either by autodiff JVP or by a single forward-difference probe. After correction, we re-anneal by sampling

$$\mathbf{x}_{t-1} = \mathbf{x}_{0|t}^{\text{corr}} + \sigma_{t-1}\,\boldsymbol{\xi}, \qquad \boldsymbol{\xi} \sim \mathcal{N}(\mathbf{0}, I), \tag{37}$$

which implements the decoupled time–marginal transport.

**Standing assumptions.**

**A1** (Noise model and metric) We assume additive white Gaussian noise (AWGN) with covariance $\beta^2 I$ and work in the standard Euclidean metric in measurement space; the feasibility set is the ball $\{\mathbf{v} : \|\mathbf{v} - \mathbf{y}\| \leq \varepsilon\}$.

**A2** (Regularity) $\mathcal{A}$ is $C^1$ in a neighborhood of the iterates, and $J_{\mathcal{A}}$ is locally Lipschitz.

**A3** (Tolerance) The radius $\varepsilon$ is treated as a user-chosen data-consistency tolerance; we do not require $\|\mathcal{A}(\mathbf{x}_0) - \mathbf{y}\| \leq \varepsilon$.

### A.3.2 PIXEL-SPACE PROPOSITIONS AND PROOFS

We restate the pixel-space results referenced in the main text and provide detailed proofs.

**Proposition 4** (Closed-form projection onto the measurement ball). *Let* $\mathcal{C} = \{\mathbf{v} \in \mathbb{R}^m : \|\mathbf{v} - \mathbf{y}\| \leq \varepsilon\}$ *in the measurement space. Then the Euclidean projection* $\Pi_{\mathcal{C}}(\mathbf{w})$ *in Equation 14 is exactly the radial shrink (*[Parikh & Boyd (2014)](#)*)*

$$\Pi_{\mathcal{C}}(\mathbf{w}) = \begin{cases} \mathbf{w}, & \|\mathbf{w} - \mathbf{y}\| \leq \varepsilon, \\ \mathbf{y} + \varepsilon\,\dfrac{\mathbf{w} - \mathbf{y}}{\|\mathbf{w} - \mathbf{y}\|}, & \text{otherwise.} \end{cases}$$

*Proof of Proposition 4.* We solve $\min_{\mathbf{v}} \frac{1}{2}\|\mathbf{v} - \mathbf{w}\|^2$ s.t. $\|\mathbf{v} - \mathbf{y}\| \leq \varepsilon$. The objective is 1-strongly convex and the feasible set is closed and convex; hence there is a unique minimizer.

**KKT derivation.** The Lagrangian is

$$\mathcal{L}(\mathbf{v}, \lambda) = \tfrac{1}{2}\|\mathbf{v} - \mathbf{w}\|^2 + \lambda\big(\|\mathbf{v} - \mathbf{y}\| - \varepsilon\big), \qquad \lambda \geq 0.$$

Stationarity gives

$$\mathbf{0} = \nabla_{\mathbf{v}}\mathcal{L}(\mathbf{v}, \lambda) = (\mathbf{v} - \mathbf{w}) + \lambda\,\frac{\mathbf{v} - \mathbf{y}}{\|\mathbf{v} - \mathbf{y}\|} \quad \text{if } \mathbf{v} \neq \mathbf{y}.$$

There are two cases.

(i) Interior case. If the constraint is inactive at the optimum, then $\lambda = 0$ by complementary slackness and stationarity gives $\mathbf{v} = \mathbf{w}$. Feasibility requires $\|\mathbf{w} - \mathbf{y}\| \leq \varepsilon$, i.e., $\mathbf{w} \in \mathcal{C}$.

(ii) Boundary case. Otherwise $\|\mathbf{v} - \mathbf{y}\| = \varepsilon$ and $\lambda > 0$. Stationarity implies $\mathbf{v} - \mathbf{w}$ is colinear with $\mathbf{v} - \mathbf{y}$; hence the optimizer lies on the ray from $\mathbf{y}$ to $\mathbf{w}$. Write $\mathbf{v} = \mathbf{y} + \tau(\mathbf{w} - \mathbf{y})$ with $\tau \geq 0$. Enforcing $\|\mathbf{v} - \mathbf{y}\| = \varepsilon$ yields $\tau = \varepsilon/\|\mathbf{w} - \mathbf{y}\|$. Substituting gives

$$\mathbf{v} = \mathbf{y} + \varepsilon\,\frac{\mathbf{w} - \mathbf{y}}{\|\mathbf{w} - \mathbf{y}\|}.$$

This is exactly the radial projection formula in Equation 16. Uniqueness follows from strong convexity. $\qquad\square$

**Proposition 1** (Local model-optimal step and descent). *Under $C^1$ regularity of $\mathcal{A}$ near $\mathbf{x}$ and local Lipschitzness of $J_{\mathcal{A}}$, if $\mathbf{g} \neq \mathbf{0}$ then $\alpha^\star$ in Equation 21 minimizes the quadratic model $\tilde{F}$ in Equation 20. Moreover, using $\alpha^\star$ to initialize an Armijo backtracking line search guarantees that the accepted step yields monotone decrease of the current $\mathbf{x}$-subproblem objective $F$ in Equation 17 (with $\mathbf{b}^k$ fixed), even when Equation 19 is only locally accurate. If $\mathbf{g} = \mathbf{0}$, then $\mathbf{x}$ is stationary for $F$ and no step is taken.*

*Proof of Proposition 1.* Write $F(\mathbf{x}) = \frac{1}{2\gamma_t}\|\mathbf{s}\|^2 + \frac{\rho}{2}\|\mathbf{r}\|^2$ with $\mathbf{s} = \mathbf{x} - \mathbf{x}_{0|t}$ and $\mathbf{r} = \mathcal{A}(\mathbf{x}) - \mathbf{b}$. The gradient is

$$\mathbf{g} = \nabla F(\mathbf{x}) = \tfrac{1}{\gamma_t}\mathbf{s} + \rho\, J_{\mathcal{A}}(\mathbf{x})^\top \mathbf{r}.$$

Consider the steepest-descent trial $\mathbf{x}(\alpha) = \mathbf{x} - \alpha\mathbf{g}$. A first-order Taylor expansion along $-\mathbf{g}$ gives

$$\mathcal{A}\big(\mathbf{x}(\alpha)\big) = \mathcal{A}(\mathbf{x}) - \alpha J_{\mathcal{A}}(\mathbf{x})\mathbf{g} + \mathbf{e}(\alpha), \qquad \|\mathbf{e}(\alpha)\| \leq \tfrac{L_{\mathcal{A}}}{2}\alpha^2\|\mathbf{g}\|^2,$$

for some local Lipschitz constant $L_{\mathcal{A}}$ of $J_{\mathcal{A}}$ (from **A2**). Plugging this into $F(\mathbf{x}(\alpha))$ yields

$$F(\mathbf{x}(\alpha)) = \underbrace{\tfrac{1}{2\gamma_t}\|\mathbf{s} - \alpha\mathbf{g}\|^2 + \tfrac{\rho}{2}\|\mathbf{r} - \alpha J_{\mathcal{A}}(\mathbf{x})\mathbf{g}\|^2}_{:=\tilde{F}(\alpha)} + \rho\langle \mathbf{r} - \alpha J_{\mathcal{A}}(\mathbf{x})\mathbf{g},\, \mathbf{e}(\alpha)\rangle + \tfrac{\rho}{2}\|\mathbf{e}(\alpha)\|^2.$$

The model $\tilde{F}$ is a convex quadratic in $\alpha$ with derivative

$$\tilde{F}'(\alpha) = -\tfrac{1}{\gamma_t}\langle \mathbf{s}, \mathbf{g}\rangle - \rho\langle \mathbf{r}, J_{\mathcal{A}}(\mathbf{x})\mathbf{g}\rangle + \alpha\Big(\tfrac{1}{\gamma_t}\|\mathbf{g}\|^2 + \rho\|J_{\mathcal{A}}(\mathbf{x})\mathbf{g}\|^2\Big),$$

and curvature $\tilde{F}''(\alpha) = \tfrac{1}{\gamma_t}\|\mathbf{g}\|^2 + \rho\|J_{\mathcal{A}}(\mathbf{x})\mathbf{g}\|^2 \geq 0$, with equality only at stationary points where $\mathbf{g} = \mathbf{0}$ and $J_{\mathcal{A}}(\mathbf{x})\mathbf{g} = \mathbf{0}$. Setting $\tilde{F}'(\alpha) = 0$ yields the model minimizer $\alpha^\star$ in Equation 21.

**Descent of the true $F$.** Using the expansion above and Cauchy–Schwarz with the bound on $\|\mathbf{e}(\alpha)\|$, we obtain

$$F(\mathbf{x} - \alpha\mathbf{g}) \leq \tilde{F}(\alpha) + \rho\|\mathbf{r} - \alpha J_{\mathcal{A}}(\mathbf{x})\mathbf{g}\|\,\tfrac{L_{\mathcal{A}}}{2}\alpha^2\|\mathbf{g}\|^2 + \tfrac{\rho}{2}\Big(\tfrac{L_{\mathcal{A}}}{2}\alpha^2\|\mathbf{g}\|^2\Big)^2.$$

At $\alpha = \alpha^\star$, $\tilde{F}(\alpha^\star) = \min_\alpha \tilde{F}(\alpha)$ and the improvement over $\tilde{F}(0) = F(\mathbf{x})$ is

$$\tilde{F}(0) - \tilde{F}(\alpha^\star) = \frac{\big(\tfrac{1}{\gamma_t}\langle \mathbf{s}, \mathbf{g}\rangle + \rho\langle \mathbf{r}, J_{\mathcal{A}}(\mathbf{x})\mathbf{g}\rangle\big)^2}{2\big(\tfrac{1}{\gamma_t}\|\mathbf{g}\|^2 + \rho\|J_{\mathcal{A}}(\mathbf{x})\mathbf{g}\|^2\big)}.$$

The remainder terms are $O(\alpha^{*2}\|\mathbf{g}\|^2)$ and $O(\alpha^{*4}\|\mathbf{g}\|^4)$; shrinking $\alpha$ by a constant factor (standard Armijo backtracking) ensures these are dominated by the quadratic-model decrease, yielding strict descent of $F$. $\qquad\square$

**Remark 4** (Step size from finite-difference JVP). *Replacing $J_{\mathcal{A}}(\mathbf{x})\mathbf{g}$ in Equation 21 by $\Delta\mathcal{A}/\eta$ from Equation 22 yields the numerically stable single-forward-call step*

$$\alpha_{\text{FD}} = \frac{\eta^2\,\tfrac{1}{\gamma_t}\langle \mathbf{s},\, \mathbf{g}\rangle\, +\, \eta\,\rho\,\langle \mathbf{r},\, \Delta\mathcal{A}\rangle}{\eta^2\,\tfrac{1}{\gamma_t}\|\mathbf{g}\|^2\, +\, \rho\,\|\Delta\mathcal{A}\|^2} \qquad where \quad \Delta\mathcal{A} = \mathcal{A}(\mathbf{x} + \eta\mathbf{g}) - \mathcal{A}(\mathbf{x}). \tag{38}$$

*which is algebraically equivalent to substituting $J_{\mathcal{A}}(\mathbf{x})\mathbf{g} \approx \Delta\mathcal{A}/\eta$ in Equation 21 (the scaling by $\eta^2$ avoids division by small $\eta$). Since $J_{\mathcal{A}}$ is locally Lipschitz, $\Delta\mathcal{A}/\eta = J_{\mathcal{A}}(\mathbf{x})\mathbf{g} + O(\eta\|\mathbf{g}\|^2)$, so $\alpha_{\text{FD}} \to \alpha^\star$ as $\eta \to 0$; backtracking preserves monotone decrease of $F$.*

**Remark 5** (Linear $\mathcal{A}$ yields exact optimal line search). *If $\mathcal{A}$ is linear, then Equation 19 is exact and Equation 21 gives the true optimal line-search step for $F$ along $-\mathbf{g}$ (Nocedal & Wright (2006)), delivering the fastest progress among steepest-descent steps.*

**Justification.** If $\mathcal{A}(\mathbf{x}) = H\mathbf{x}$, then $J_{\mathcal{A}}(\mathbf{x}) = H$ and the linearization is exact: $\mathcal{A}(\mathbf{x} - \alpha\mathbf{g}) = \mathcal{A}(\mathbf{x}) - \alpha H\mathbf{g}$. Hence $\tilde{F}$ coincides with $F(\mathbf{x} - \alpha\mathbf{g})$ along the line, and the model minimizer in Equation 21 is the exact optimal line-search step.

**Proposition 5** (Fixed points satisfy KKT for Equation 9). *Let $(\mathbf{x}^*, \mathbf{v}^*, \mathbf{u}^*)$ be a fixed point of the exact scaled-ADMM updates Equation 13–Equation 15 (i.e., $\mathbf{x}^{k+1}$ solves the $\mathbf{x}$-subproblem Equation 13 exactly). Then $\mathcal{A}(\mathbf{x}^*) = \mathbf{v}^*$, $\mathbf{v}^* \in \mathcal{C}$, and there exists $\lambda^* \geq 0$ such that*

$$\frac{1}{\gamma_t}(\mathbf{x}^* - \mathbf{x}_{0|t}) + \lambda^* J_{\mathcal{A}}(\mathbf{x}^*)^\top \boldsymbol{\nu}^* = 0, \qquad \lambda^*\big(\|\mathcal{A}(\mathbf{x}^*) - \mathbf{y}\| - \varepsilon\big) = 0, \qquad (39)$$

*where*

$$\boldsymbol{\nu}^* \in \begin{cases} \left\{ \dfrac{\mathcal{A}(\mathbf{x}^*) - \mathbf{y}}{\|\mathcal{A}(\mathbf{x}^*) - \mathbf{y}\|} \right\}, & \|\mathcal{A}(\mathbf{x}^*) - \mathbf{y}\| = \varepsilon, \\ \{\mathbf{0}\}, & \|\mathcal{A}(\mathbf{x}^*) - \mathbf{y}\| < \varepsilon. \end{cases}$$

*Hence $\mathbf{x}^*$ satisfies the KKT conditions of Equation 9 ([Bertsekas (1999)](#)).*

*Proof of Proposition 5.* At a fixed point $(\mathbf{x}^*, \mathbf{v}^*, \mathbf{u}^*)$, the $\mathbf{u}$-update satisfies $\mathbf{u}^* = \mathbf{u}^* + \mathcal{A}(\mathbf{x}^*) - \mathbf{v}^*$, hence primal feasibility $\mathcal{A}(\mathbf{x}^*) - \mathbf{v}^* = \mathbf{0}$. The $\mathbf{v}$-update is the metric projection onto $\mathcal{C}$:

$$\mathbf{v}^* = \Pi_{\mathcal{C}}(\mathcal{A}(\mathbf{x}^*) + \mathbf{u}^*),$$

so $\mathbf{v}^* \in \mathcal{C}$ and the optimality condition of the projection reads

$$\mathbf{0} \in \partial \iota_{\mathcal{C}}(\mathbf{v}^*) + \rho\big(\mathbf{v}^* - (\mathcal{A}(\mathbf{x}^*) + \mathbf{u}^*)\big) = \partial \iota_{\mathcal{C}}(\mathbf{v}^*) - \rho\,\mathbf{u}^*,$$

i.e., $\rho\,\mathbf{u}^* \in \partial \iota_{\mathcal{C}}(\mathbf{v}^*) = N_{\mathcal{C}}(\mathbf{v}^*)$, the normal cone of $\mathcal{C}$ at $\mathbf{v}^*$. For the $\mathbf{x}$-subproblem, first-order optimality gives

$$\mathbf{0} = \tfrac{1}{\gamma_t}(\mathbf{x}^* - \mathbf{x}_{0|t}) + \rho\, J_{\mathcal{A}}(\mathbf{x}^*)^\top\big(\mathcal{A}(\mathbf{x}^*) - \mathbf{v}^* + \mathbf{u}^*\big) = \tfrac{1}{\gamma_t}(\mathbf{x}^* - \mathbf{x}_{0|t}) + \rho\, J_{\mathcal{A}}(\mathbf{x}^*)^\top \mathbf{u}^*,$$

using primal feasibility. The normal cone for the ball $\mathcal{C} = \{\mathbf{v} : \|\mathbf{v} - \mathbf{y}\| \leq \varepsilon\}$ is

$$N_{\mathcal{C}}(\mathbf{v}^*) = \begin{cases} \{\lambda\boldsymbol{\nu}^* : \lambda \geq 0\}, & \|\mathbf{v}^* - \mathbf{y}\| = \varepsilon, \\ \{\mathbf{0}\}, & \|\mathbf{v}^* - \mathbf{y}\| < \varepsilon, \end{cases} \quad \text{with} \quad \boldsymbol{\nu}^* = \frac{\mathbf{v}^* - \mathbf{y}}{\|\mathbf{v}^* - \mathbf{y}\|}.$$

Thus $\rho\,\mathbf{u}^* = \lambda^*\boldsymbol{\nu}^*$ for some $\lambda^* \geq 0$ when the constraint is active and $\mathbf{u}^* = \mathbf{0}$ otherwise. Substituting into the $\mathbf{x}$-optimality condition yields

$$\tfrac{1}{\gamma_t}(\mathbf{x}^* - \mathbf{x}_{0|t}) + \lambda^* J_{\mathcal{A}}(\mathbf{x}^*)^\top \boldsymbol{\nu}^* = \mathbf{0}.$$

Complementarity $\lambda^*(\|\mathcal{A}(\mathbf{x}^*) - \mathbf{y}\| - \varepsilon) = 0$ follows by construction of the normal cone. Hence $(\mathbf{x}^*, \lambda^*)$ satisfies the KKT conditions of Equation 33. $\qquad\square$

**Proposition 6** (Mode-substitution re-annealing under a local Gaussian surrogate). *Fix $\mathbf{x}_t$ and assume the conditional admits a local Gaussian (Laplace) surrogate $p(\mathbf{x}_0 \mid \mathbf{x}_t, \mathbf{y}) \approx \mathcal{N}(\boldsymbol{m}_t, \Sigma_t)$ with $\Sigma_t \succeq 0$, where $\boldsymbol{m}_t$ is the local mode/mean of the surrogate. Let $\mathbf{x}_{0|t}^{\mathrm{corr}}$ denote the deterministic corrected estimate used by FAST-DIPS as the center of the re-annealing step in Equation 11 (for analysis, take $\mathbf{x}_{0|t}^{\mathrm{corr}}$ to be the solution of Equation 9). For $\sigma_{t-1} > 0$, consider the two per-step Gaussian-injected distributions: (i) sample $\mathbf{x}_0 \sim \mathcal{N}(\boldsymbol{m}_t, \Sigma_t)$ and inject $\mathcal{N}(\mathbf{0}, \sigma_{t-1}^2 I)$, giving $\mathcal{N}(\boldsymbol{m}_t, \Sigma_t + \sigma_{t-1}^2 I)$; versus (ii) inject $\mathcal{N}(\mathbf{0}, \sigma_{t-1}^2 I)$ centered at $\mathbf{x}_{0|t}^{\mathrm{corr}}$, giving $\mathcal{N}(\mathbf{x}_{0|t}^{\mathrm{corr}}, \sigma_{t-1}^2 I)$. Then*

$$\mathrm{KL}\Big(\mathcal{N}(\boldsymbol{m}_t, \Sigma_t + \sigma_{t-1}^2 I) \,\big\|\, \mathcal{N}(\mathbf{x}_{0|t}^{\mathrm{corr}}, \sigma_{t-1}^2 I)\Big) \leq \frac{\|\boldsymbol{m}_t - \mathbf{x}_{0|t}^{\mathrm{corr}}\|^2}{2\sigma_{t-1}^2} + \frac{\|\Sigma_t\|_F^2}{4\sigma_{t-1}^4}. \qquad (40)$$

**Consequences.** *The bound quantifies the approximation incurred by FAST-DIPS replacing sampling from the local Gaussian surrogate by mode-centered re-annealing. It is small early when $\sigma_{t-1}^2$ is large, and it is small late when the surrogate conditional is concentrated (small $\|\Sigma_t\|_F$) and the corrected estimate $\mathbf{x}_{0|t}^{\mathrm{corr}}$ is close to the surrogate mode/mean $\boldsymbol{m}_t$. The first term also makes explicit that an arbitrary substitution can be poor; the approximation is meaningful only insofar as $\mathbf{x}_{0|t}^{\mathrm{corr}}$ acts as a good mode proxy under the local surrogate.*

*Proof of Proposition 6.* Let $P = \mathcal{N}(\boldsymbol{m}_t, \Sigma_t + \sigma^2 I)$ and $Q = \mathcal{N}(\mathbf{x}_{0|t}^{\text{corr}}, \sigma^2 I)$ in $\mathbb{R}^d$. The Gaussian KL formula gives

$$\text{KL}(P\|Q) = \tfrac{1}{2}\left(\text{tr}(\Sigma_Q^{-1}\Sigma_P) + (\boldsymbol{\mu}_Q - \boldsymbol{\mu}_P)^\top \Sigma_Q^{-1}(\boldsymbol{\mu}_Q - \boldsymbol{\mu}_P) - d + \log\frac{\det\Sigma_Q}{\det\Sigma_P}\right).$$

With $\Sigma_Q = \sigma^2 I$, $\Sigma_P = \sigma^2 I + \Sigma_t$, $\boldsymbol{\mu}_Q - \boldsymbol{\mu}_P = \mathbf{x}_{0|t}^{\text{corr}} - \boldsymbol{m}_t$, we get

$$\text{KL}(P\|Q) = \frac{\|\mathbf{x}_{0|t}^{\text{corr}} - \boldsymbol{m}_t\|^2}{2\sigma^2} + \frac{1}{2}\left(\text{tr}(I + \tfrac{1}{\sigma^2}\Sigma_t) - d - \log\det(I + \tfrac{1}{\sigma^2}\Sigma_t)\right).$$

Diagonalize $\Sigma_t = U\Lambda U^\top$ with eigenvalues $\lambda_i \geq 0$. Then

$$\text{KL}(P\|Q) = \frac{\|\mathbf{x}_{0|t}^{\text{corr}} - \boldsymbol{m}_t\|^2}{2\sigma^2} + \frac{1}{2}\sum_{i=1}^d \left(\frac{\lambda_i}{\sigma^2} - \log\left(1 + \frac{\lambda_i}{\sigma^2}\right)\right).$$

Use $x - \log(1 + x) \leq x^2/2$ for $x \geq 0$ termwise to obtain

$$\text{KL}(P\|Q) \leq \frac{\|\mathbf{x}_{0|t}^{\text{corr}} - \boldsymbol{m}_t\|^2}{2\sigma^2} + \frac{1}{4}\sum_{i=1}^d \frac{\lambda_i^2}{\sigma^4} = \frac{\|\mathbf{x}_{0|t}^{\text{corr}} - \boldsymbol{m}_t\|^2}{2\sigma^2} + \frac{\|\Sigma_t\|_F^2}{4\sigma^4}.$$

**Tightness regimes.** The second term vanishes as $\sigma^2 \to \infty$ (early in the schedule) and as $\|\Sigma_t\|_F \to 0$ (late in the schedule); the first term quantifies bias between the mode $\mathbf{x}_{0|t}^{\text{corr}}$ and the posterior mean $\boldsymbol{m}_t$. $\square$

### A.3.3 LATENT-SPACE COUNTERPARTS

**Why the substitution $\mathcal{A} \mapsto \mathcal{A}\circ\mathcal{D}$ is valid.** If $\mathcal{A}$ and the decoder $\mathcal{D}$ are $C^1$ in a neighborhood of the iterates and have locally Lipschitz Jacobians, then the composite $\mathcal{A}\circ\mathcal{D}$ is $C^1$ with locally Lipschitz Jacobian on the same neighborhood. Consequently, all arguments that rely on local linearization (and VJP/JVP computations) transfer verbatim to $\mathcal{A}\circ\mathcal{D}$ via the chain rule, while the projection step is unchanged because feasibility is enforced in measurement space.

**Latent proofs.** The latent descent guarantee and the latent fixed-point $\Rightarrow$ KKT statement follow from the pixel-space proofs by replacing $\mathcal{A}$ with $\mathcal{A}\circ\mathcal{D}$ and $\mathbf{x}$ with $\mathbf{z}$, and applying the chain rule to the VJP/JVP terms. Since these arguments are structurally identical, we omit the redundant proofs for brevity.

**Remark 6** (Mode-substitution transport in latent space). *Proposition 6 applies verbatim to the latent re-annealing step $\mathbf{z}_{t-1} = \mathbf{z}_{0|t}^{\text{corr}} + \sigma_{t-1}\boldsymbol{\xi}$ by replacing $\mathbf{x}$ with $\mathbf{z}$, since the transport kernel in latent space is Gaussian. If one compares the decoded distributions in pixel space, deterministic decoding $\mathcal{D}$ cannot increase KL (data processing), so the latent KL bound also upper-bounds the discrepancy after decoding.*

### A.3.4 DERIVATION OF ANALYTIC STEP SIZES AND AUTODIFF COMPUTATION

**Pixel space:** Recall

$$F(\mathbf{x}) = \frac{1}{2\gamma_t}\|\mathbf{x} - \mathbf{x}_{0|t}\|^2 + \frac{\rho}{2}\|\mathcal{A}(\mathbf{x}) - \mathbf{b}\|^2, \quad \mathbf{s} = \mathbf{x} - \mathbf{x}_{0|t}, \quad \mathbf{r} = \mathcal{A}(\mathbf{x}) - \mathbf{b}.$$

Then $\mathbf{g} = \frac{1}{\gamma_t}\mathbf{s} + \rho\, J_\mathcal{A}(\mathbf{x})^\top \mathbf{r}$. For the trial $\mathbf{x}(\alpha) = \mathbf{x} - \alpha\mathbf{g}$,

$$\mathcal{A}(\mathbf{x}(\alpha)) \approx \mathcal{A}(\mathbf{x}) - \alpha J_\mathcal{A}(\mathbf{x})\mathbf{g}$$

gives the scalar quadratic model

$$\tilde{F}(\alpha) = \frac{1}{2\gamma_t}\|\mathbf{s} - \alpha\mathbf{g}\|^2 + \frac{\rho}{2}\|\mathbf{r} - \alpha J_\mathcal{A}(\mathbf{x})\mathbf{g}\|^2,$$

whose derivative is

$$\tilde{F}'(\alpha) = -\frac{1}{\gamma_t}\langle \mathbf{s}, \mathbf{g}\rangle - \rho\langle \mathbf{r}, J_{\mathcal{A}}(\mathbf{x})\mathbf{g}\rangle + \alpha\left(\frac{1}{\gamma_t}\|\mathbf{g}\|^2 + \rho\|J_{\mathcal{A}}(\mathbf{x})\mathbf{g}\|^2\right).$$

Setting $\tilde{F}'(\alpha) = 0$ yields $\alpha^\star$ in Equation 21. The curvature $\tilde{F}''(\alpha) = \frac{1}{\gamma_t}\|\mathbf{g}\|^2 + \rho\|J_{\mathcal{A}}(\mathbf{x})\mathbf{g}\|^2 \geq 0$ shows uniqueness unless $\mathbf{g} = \mathbf{0}$.

*Autodiff computation recipe (pixel):*

1. Evaluate $\mathcal{A}(\mathbf{x})$ to get $\mathbf{r} = \mathcal{A}(\mathbf{x}) - \mathbf{b}$.

2. Compute the VJP $J_{\mathcal{A}}(\mathbf{x})^\top \mathbf{r}$ (reverse-mode autodiff) and form $\mathbf{g}$.

3. Obtain the directional Jacobian $J_{\mathcal{A}}(\mathbf{x})\mathbf{g}$ either

   - by forward-mode autodiff (preferred when available), or
   - by a single forward-difference probe

   $$J_{\mathcal{A}}(\mathbf{x})\mathbf{g} \approx \frac{\Delta\mathcal{A}}{\eta}, \qquad \Delta\mathcal{A} := \mathcal{A}(\mathbf{x} + \eta\mathbf{g}) - \mathcal{A}(\mathbf{x}),$$

   in which case it is numerically convenient to assemble the FD-stabilized closed form Equation 38 (equivalent to substituting $\Delta\mathcal{A}/\eta$ into Equation 21 but avoiding division by small $\eta$).

4. Assemble the numerator/denominator and perform Armijo backtracking.

**Latent space:** With

$$F_z(\mathbf{z}) = \frac{1}{2\gamma_z}\|\mathbf{z} - \mathbf{z}_{0|t}\|^2 + \frac{\rho_z}{2}\|\mathcal{A}(\mathcal{D}(\mathbf{z})) - \mathbf{b}\|^2, \quad \mathbf{g}_z = \frac{1}{\gamma_z}(\mathbf{z} - \mathbf{z}_{0|t}) + \rho_z\, J_{\mathcal{A}\circ\mathcal{D}}(\mathbf{z})^\top(\mathcal{A}(\mathcal{D}(\mathbf{z})) - \mathbf{b}),$$

linearize $\mathcal{A}\circ\mathcal{D}$ to obtain

$$\tilde{F}_z(\alpha) = \frac{1}{2\gamma_z}\|\mathbf{z} - \mathbf{z}_{0|t} - \alpha\mathbf{g}_z\|^2 + \frac{\rho_z}{2}\|\mathcal{A}(\mathcal{D}(\mathbf{z})) - \mathbf{b} - \alpha\, J_{\mathcal{A}\circ\mathcal{D}}(\mathbf{z})\mathbf{g}_z\|^2,$$

whose minimizer is Equation 32. The VJP/JVP are computed end-to-end through $\mathcal{D}$ and $\mathcal{A}$ by autodiff; a single finite-difference through the composition is a valid JVP fallback:

$$J_{\mathcal{A}\circ\mathcal{D}}(\mathbf{z})\mathbf{g}_z \approx \frac{\mathcal{A}(\mathcal{D}(\mathbf{z} + \delta\mathbf{g}_z)) - \mathcal{A}(\mathcal{D}(\mathbf{z}))}{\delta}.$$

**Complex-valued measurements.** When measurements are complex, we work with real–imaginary stacking (dimension $2m$) and the Euclidean norm; all expressions remain valid verbatim, with $J_{\mathcal{A}}$ denoting the real Jacobian.

**Backtracking and safeguards.** We use a monotone backtracking line search: starting from an initial candidate step (e.g., $\alpha^\star$ or $\alpha_{\mathrm{FD}}$), repeatedly shrink $\alpha \leftarrow \tau\alpha$ (e.g., $\tau = \frac{1}{2}$) until acceptance, i.e.,

$$F(\mathbf{x} - \alpha\mathbf{g}) \leq F(\mathbf{x}) \quad \text{(up to a small numerical tolerance)}.$$

### A.3.5 ADDITIONAL REMARKS

**Remark 7** (Trust-region scaling along the schedule). *Setting $\gamma_t = \sigma_t^2$ ties the proximal weight to the diffusion noise: early (large $\sigma_t$) the correction can move farther from the denoiser anchor, while late (small $\sigma_t$) the local trust region tightens.*

**Remark 8** (Constraint and closed-form projection in implementation). *Under the AWGN setting adopted throughout, the measurement-space constraint is an $\ell_2$ (Euclidean) ball and the $\mathbf{v}$-update is the closed-form radial shrink in Equation 16. The remaining steps follow standard scaled ADMM-style updates, with the $\mathbf{x}$-update handled approximately by adjoint-free gradient steps with analytic step sizing and backtracking.*

Table 2: Ablation of step-size selection inside two per-level solvers. Left: Gaussian blur (pixel). Right: HDR (latent). We compare constant $\alpha$, analytic/JVP $\alpha^*$, and forward-only $\alpha_{\text{FD}}$ within QDP (no splitting, no proj.) and FAST-DIPS (ADMM + proj.). For fairness, compute is matched by allocating $K \times S$ gradient steps per level to QDP when FAST-DIPS uses $K$ ADMM iterations with $S$ gradient steps each; projection/dual updates are negligible. All results are evaluated on FFHQ256 10 samples.

| Solver | Step Size Method | | Gaussian Blur (Pixel) | | | | Solver | Step Size Method | | High Dynamic Range (Latent) | | | |
|---|---|---|---|---|---|---|---|---|---|---|---|---|---|
| | | | PSNR | SSIM | LPIPS | Run-time (s) | | | | PSNR | SSIM | LPIPS | Run-time (s) |
| QDP (no splitting, no proj.) | constant | $\alpha = 10^{-4}$ | 22.854 | 0.665 | 0.429 | 1.893 | QDP (no splitting, no proj.) | constant | $\alpha = 10^{-5}$ | 22.113 | 0.671 | 0.459 | 21.827 |
| | | $\alpha = 10^{-3}$ | 28.028 | 0.796 | 0.314 | 1.867 | | | $\alpha = 10^{-4}$ | 23.486 | 0.769 | 0.356 | 22.078 |
| | | $\alpha = 10^{-2}$ | 2.687 | 0.162 | 0.779 | 1.955 | | | $\alpha = 10^{-3}$ | 16.296 | 0.614 | 0.555 | 22.044 |
| | JVP | | 29.480 | 0.830 | 0.271 | 2.356 | | JVP | | 24.356 | 0.757 | 0.357 | 60.963 |
| | FD | | 29.577 | 0.832 | 0.268 | 2.018 | | FD | | 23.196 | 0.750 | 0.364 | 31.158 |
| FAST-DIPS (ADMM + proj.) | constant | $\alpha = 10^{-4}$ | 24.829 | 0.714 | 0.391 | 1.988 | FAST-DIPS (ADMM + proj.) | constant | $\alpha = 10^{-6}$ | 21.522 | 0.641 | 0.496 | 25.011 |
| | | $\alpha = 10^{-3}$ | 28.647 | 0.811 | 0.296 | 2.029 | | | $\alpha = 10^{-5}$ | 25.021 | 0.768 | 0.339 | 25.110 |
| | | $\alpha = 10^{-2}$ | 3.851 | 0.151 | 0.772 | 1.993 | | | $\alpha = 10^{-4}$ | 23.328 | 0.797 | 0.298 | 25.159 |
| | JVP | | 29.762 | 0.829 | 0.273 | 2.502 | | JVP (Ours) | | 25.530 | 0.811 | 0.273 | 63.952 |
| | FD (Ours) | | 29.632 | 0.819 | 0.287 | 2.053 | | FD | | 21.041 | 0.736 | 0.355 | 34.197 |

**Remark 9** (Empirical choice: FD in pixel, JVP in latent). *Our analytic step size depends on a directional Jacobian term along the current descent direction: $J_{\mathcal{A}}(\mathbf{x})\mathbf{g}$ in pixel space and $J_{\mathcal{A} \circ \mathcal{D}}(\mathbf{z})\mathbf{g}_z$ in latent space. In pixel space, a single forward-difference probe provides a sufficiently accurate estimate of this term at lower overhead than an autodiff JVP in our implementation, yielding similar performance with reduced run-time. In latent space, however, the composed map $\mathcal{A} \circ \mathcal{D}$ is typically much more curved, so the FD estimate becomes sensitive to the perturbation scale: too large yields a biased directional derivative (poor local linearization), while too small suffers from numerical cancellation (especially with mixed precision), leading to unstable step-size estimates and degraded reconstructions. We therefore use autodiff JVPs through $\mathcal{A} \circ \mathcal{D}$ for the latent variant. In both cases, backtracking guarantees descent of the current subproblem objective (with ADMM auxiliary variables fixed).*

## A.4 ABLATION STUDIES

**Goal and tasks.** We assess the impact of measurement-space feasibility via projection and the choice of step size inside the $\mathbf{x}$-update. Experiments use 10 FFHQ images on two representatives: Gaussian blur in pixel space and HDR in latent space, with PSNR/SSIM/LPIPS and average per-image run-time.

**Baseline and objective.** To isolate projection, we evaluate an unsplit penalized baseline that optimizes the same quadratic objective as the $\mathbf{x}$-subproblem inside ADMM, but without variable splitting or projection:

$$\min_{\mathbf{x} \in \mathbb{R}^{CHW}} \frac{1}{2\gamma_t} \|\mathbf{x} - \mathbf{x}_{0|t}\|^2 + \frac{1}{2\beta^2} \|\mathcal{A}(\mathbf{x}) - \mathbf{y}\|^2,$$

which we refer to as QDP (no splitting, no proj.). In all runs we match the ADMM instantiation by setting $\gamma_t = \sigma_t^2$ identically to FAST-DIPS and choosing the data-penalty weight so that $\frac{\rho}{2} = \frac{1}{2\beta^2}$.

**Compute-matched fairness.** Each $\mathbf{x}$-gradient step entails one forward pass of $\mathcal{A}$, one VJP, and one JVP (or a single forward probe for FD); projection and dual updates are negligible. With $K$ ADMM iterations and $S$ gradient steps per iteration, FAST-DIPS (ADMM + proj.) spends $K \times S$ such triplets per level, so QDP is allotted $K \times S$ gradient steps per level to match compute. Step-size mechanisms are kept identical between solvers: constant $\alpha$, analytic/JVP $\alpha^\star$, and finite-difference $\alpha_{\text{FD}}$.

**Findings.** As shown in Table 2, in pixel space, $\alpha_{\text{FD}}$ is competitive with $\alpha^\star$ at lower cost; in latent space, $\alpha^\star$ provides the stability needed for the nonlinear decoder–forward composition, while $\alpha_{\text{FD}}$ lags. Under the matched budget, enforcing feasibility via projection improves quality over the unsplit penalty path; latent run-times primarily reflect decoder backprop.

Table 3: The hyperparameters of experiments in paper for all tasks.

| Algorithm | Parameter | Super Resolution 4× | Inpaint (Box) | Linear task Inpaint (Random) | Gaussian deblurring | Motion deblurring | Phase retrieval | Non Linear task Nonlinear deblurring | High dynamic range |
|---|---|---|---|---|---|---|---|---|---|
| FAST-DIPS | $T$ | 75 | 75 | 75 | 50 | 50 | 150 | 150 | 150 |
| | $K$ | 3 | 3 | 3 | 3 | 3 | 2 | 2 | 2 |
| | $S$ | 1 | 1 | 1 | 2 | 2 | 5 | 5 | 5 |
| | $\rho$ | 200 | 200 | 200 | 200 | 200 | 200 | 200 | 5 |
| | $\varepsilon$ | 0.05 | 0.05 | 0.05 | 0.05 | 0.05 | 0.05 | 0.05 | 0.05 |
| Latent FAST-DIPS | $T$ | 50 | 50 | 50 | 50 | 50 | 25 | 25 | 25 |
| | $(K_x, K_z)$ | (5,5) | (5,5) | (5,5) | (5,5) | (5,5) | (10,10) | (10,10) | (10,10) |
| | $(S_x, S_z)$ | (3,3) | (3,3) | (3,3) | (3,3) | (3,3) | (3,3) | (3,3) | (3,3) |
| | $(\rho_x, \rho_z)$ | (200,200) | (200,200) | (200,200) | (200,200) | (200,200) | (200,200) | (200,200) | (200,200) |
| | $\sigma_{\text{switch}}$ | 1 | 1 | 1 | 1 | 1 | 5 | 5 | 5 |
| | $\varepsilon$ | 0.05 | 0.05 | 0.05 | 0.05 | 0.05 | 0.05 | 0.05 | 0.05 |

## A.5 HYPERPARAMETERS OVERVIEW

Throughout our experiments, hyperparameter settings are summarized in Table 3. We fix $\eta = 10^{-3}$ for all experiments. In the annealing process, we set $\sigma_{\text{max}} = 100$ in pixel space and 10 in latent space, with $\sigma_{\text{min}} = 0.1$ in both, to enhance robustness to measurement noise.

## A.6 EXPERIMENTAL DETAILS

**Validation set information.** For reproducibility, we explicitly specify the indices of the samples used for validation. **FFHQ** ($256 \times 256$). We use 100 images corresponding to dataset indices 00000–00099. **ImageNet** ($256 \times 256$). We use 100 images from the ImageNet validation set corresponding to indices 49000–49099.

**run-time measurement.** The run-times reported in Table 1 and Figure 3 correspond to the pure sampling time per generated sample. Since we sample with batch size 1, we measure the sampling time for each individual sample, sum over $N$ samples, and report the average run-time as $\left( \sum_{j=1}^{N} t_j \right) / N$. We exclude data loading, saving, and metric evaluation from the timing, and measure only the time spent inside the sampling loop. The same timing protocol is applied to FAST-DIPS and all baselines. For phase retrieval (best-of-4), we report the mean per-run run-time over four independent runs: $\left( \sum_{i=1}^{4} \tau_i \right) / 4$.

## A.7 BASELINE IMPLEMENTATION DETAILS

All baselines were run using the authors' public repositories:

- **DAPS/LatentDAPS**: github.com/zhangbingliang2019/DAPS
- **SITCOM**: github.com/sjames40/SITCOM
- **HRDIS**: github.com/deng-ai-lab/HRDIS
- **C-ΠGDM**: github.com/mandt-lab/c-pigdm
- **PSLD**: github.com/LituRout/PSLD
- **ReSample**: github.com/soominkwon/resample

We followed each method's original paper and default repository settings. Additionally, for phase retrieval we applied a best-of-four protocol uniformly across all compared baselines.

**Measurement noise setting.** Because the SVD-based operator becomes unstable when noise is injected in super-resolution and Gaussian deblurring, we evaluate HRDIS with noise on the remaining tasks, whereas C-ΠGDM is evaluated only in the noiseless setting across all tasks.

**Details of Figure 3.** For the run-time–quality trade-off in Figure 3, we varied only the number of solver steps/iterations per method, keeping all other hyperparameters at their recommended defaults:

- **DAPS:** The number of ODE steps was fixed at 4, while the number of annealing steps was swept over $\{2, 5, 10, 15, 20, 25\}$.

Figure 4: Qualitative reconstructions under Poisson measurement noise ($\lambda_{\text{poisson}} = 1$): FAST-DIPS preserves edges and textures across tasks.

Table 4: Quantitative results under Poisson measurement noise ($\lambda_{\text{poisson}} = 1$). FAST-DIPS remains accurate and perceptually faithful across tasks.

| Task | PSNR | SSIM | LPIPS |
|------|------|------|-------|
| Gaussian deblurring | 28.730 | 0.814 | 0.273 |
| Random Inpainting | 30.806 | 0.878 | 0.192 |
| Nonlinear deblurring | 27.016 | 0.781 | 0.266 |

- **SITCOM:** We swept pairs of diffusion steps $N$ and inner iterations $K$ over $(N, K) \in \{(3, 2), (5, 3), (5, 5), (5, 10), (5, 15), (5, 20)\}$.
- **HRDIS:** We varied the number of diffusion steps over $\{10, 15, 50, 80, 100, 130\}$.
- **C-ΠGDM:** We varied the number of diffusion steps over $\{20, 50, 75, 100, 150, 200\}$.

**Automatic Differentiation Primitives.** To implement the adjoint-free analytic updates without manually deriving gradients for the forward operator $\mathcal{A}$, we leverage the automatic differentiation capabilities of PyTorch. Specifically, the Vector-Jacobian Product (VJP) term $J_{\mathcal{A}}(\mathbf{x})^{\top}\mathbf{r}$, which is necessary for computing the gradient of the data-consistency term, is obtained via standard reverse-mode differentiation using `torch.autograd.grad`. For the Jacobian-Vector Product (JVP) term $J_{\mathcal{A}}(\mathbf{x})\mathbf{g}$, we utilize the functional transformation API, specifically `torch.func.jvp`. This allows us to efficiently compute the directional derivative required for the local quadratic model in a fully differentiable manner.

## A.8 ADDITIONAL EXPERIMENTS

Table 5: The trade-off between quality and cost in the x-update step. For complex nonlinear tasks such as nonlinear deblurring, increasing the number of gradient steps improves reconstruction quality but also increases computational cost. All experiments were conducted on 10 samples using an RTX 6000 Ada GPU.

| $K$ | $S$ | Super Resolution 4× | | | | Nonlinear Blur | | | |
|-----|-----|------|------|-------|--------------|------|------|-------|--------------|
| | | PSNR | SSIM | LPIPS | Run-time (s) | PSNR | SSIM | LPIPS | Run-time (s) |
| 2 | 1 | 29.627 | 0.837 | 0.260 | 1.444 | 25.101 | 0.711 | 0.367 | 8.711 |
| | 3 | 29.675 | 0.838 | 0.259 | 1.778 | 27.275 | 0.780 | 0.302 | 17.284 |
| | 5 | 29.678 | 0.838 | 0.259 | 2.113 | 27.914 | 0.800 | 0.285 | 26.766 |
| 3 | 1 | 29.740 | 0.839 | 0.256 | 1.569 | 25.491 | 0.725 | 0.347 | 10.937 |
| | 3 | 29.734 | 0.839 | 0.256 | 2.092 | 27.079 | 0.785 | 0.287 | 26.888 |
| | 5 | 29.734 | 0.839 | 0.256 | 2.608 | 27.546 | 0.801 | 0.274 | 38.314 |
| 5 | 1 | 29.736 | 0.838 | 0.252 | 1.794 | 25.501 | 0.734 | 0.323 | 15.707 |
| | 3 | 29.737 | 0.838 | 0.252 | 2.641 | 26.706 | 0.777 | 0.267 | 39.492 |
| | 5 | 29.737 | 0.838 | 0.252 | 3.418 | 27.006 | 0.786 | 0.265 | 63.044 |

**Effectiveness of $K$ ADMM iterations and $S$ gradient steps.** We performed an ablation study to evaluate the influence of both the number of ADMM iterations ($K$) and the number of gradient steps ($S$) in the x-update stage. We tested $K \in \{2, 3, 5\}$ and $S \in \{1, 3, 5\}$ across both linear and

Table 6: Sensitivity analysis of the main hyperparameters for Super resolution 4×, evaluated on 10 FFHQ images. The table shows the performance while sweeping the ADMM penalty $\rho$ and the constraint radius $\varepsilon$. The results demonstrate that our method is robust, with performance remaining remarkably stable across a wide range of values, which reduces the need for extensive hyperparameter tuning.

| $\rho$ | PSNR | SSIM | LPIPS | | $\varepsilon$ | PSNR | SSIM | LPIPS |
|---|---|---|---|---|---|---|---|---|
| 10 | 27.546 | 0.783 | 0.339 | | 0 | 29.739 | 0.839 | 0.255 |
| 100 | 29.614 | 0.836 | 0.267 | | 0.01 | 29.739 | 0.839 | 0.255 |
| 200 | 29.740 | 0.839 | 0.256 | | 0.05 | 29.740 | 0.839 | 0.256 |
| 500 | 29.565 | 0.828 | 0.260 | | 0.1 | 29.740 | 0.839 | 0.256 |
| 1000 | 29.363 | 0.816 | 0.276 | | 1 | 29.726 | 0.839 | 0.258 |

Table 7: Performance of the hybrid pixel–latent schedule with varying $\sigma_{\text{switch}}$ values for 4× super-resolution on 10 FFHQ images. The schedule performs correction in pixel space when $\sigma_t > \sigma_{\text{switch}}$ and in latent space otherwise. The results, measured on an RTX 6000 Ada GPU, show that a balanced approach ($\sigma_{\text{switch}} = 1.0$ for the linear task) is more effective than a purely pixel-space ($< 0.0$) or purely latent-space ($> 10.0$) correction strategy.

| $\sigma_{\text{switch}}$ | PSNR | SSIM | LPIPS | Run-time (s) |
|---|---|---|---|---|
| $< 0.0$ | 24.283 | 0.553 | 0.469 | 3.082 |
| 0.2 | 27.185 | 0.681 | 0.374 | 11.727 |
| 1 | 28.809 | 0.793 | 0.302 | 38.014 |
| 5 | 28.828 | 0.791 | 0.306 | 73.138 |
| $>10.0$ | 28.819 | 0.791 | 0.307 | 90.646 |

nonlinear tasks. As shown in Table 5, increasing $K$ consistently improves performance for both linear tasks (e.g., super-resolution) and nonlinear tasks (e.g., nonlinear deblurring), though naturally at the cost of higher computation. This suggests that the refinement offered by additional ADMM iterations is broadly beneficial regardless of task difficulty.

In contrast, the effect of increasing the number of gradient steps $S$ is task-dependent. For linear tasks, the performance gain is marginal relative to the additional run-time. However, for nonlinear tasks, the reconstruction metrics steadily improve as $S$ increases, indicating that additional gradient refinement helps the solver locate more accurate correction points in challenging settings. Overall, both $K$ and $S$ present a clear quality–cost trade-off, with $K$ providing general improvements and $S$ offering additional benefits especially for complex nonlinear problems.

**Hyperparameter Robustness.** We investigate the robustness of our method to its main hyperparameters. Table 6 shows the results for the super-resolution task when sweeping the ADMM penalty $\rho$, and the constraint radius $\varepsilon$. The performance remains stable across a wide range of values for each parameter. This highlights a key advantage of FAST-DIPS: it is not sensitive to fine-tuning and delivers strong results with default settings, enhancing its practicality and ease of use.

**Hybrid Schedule Trade-off.** In our hybrid pixel-latent framework, the $\sigma_{\text{switch}}$ parameter determines the point at which the correction process transitions from pixel space to latent space. Table 7 illustrates the resulting trade-off between performance and run-time. Performing the initial correction steps in pixel space ($\sigma_{\text{switch}} > 0$) provides a fast and effective rough update, significantly reducing the overall computation time. The subsequent switch to latent-space updates allows for more stable, fine-grained corrections that respect the generative manifold. This hybrid strategy proves highly effective, and an intermediate $\sigma_{\text{switch}}$ value offers an optimal balance between speed and reconstruction fidelity.

**Experiments with non-Gaussian noise.** Figure 4 and Table 4 evaluate FAST-DIPS under Poisson measurement noise with rate $\lambda_{\text{poisson}} = 1$, showing that our method remains accurate and perceptually faithful beyond the additive white Gaussian noise (AWGN) setting. The robustness arises from replacing a parametric likelihood with a set-valued surrogate: at each diffusion level, we solve a denoiser-anchored, hard-constrained proximal problem that enforces feasibility within a measurement-space feasibility ball. This set-valued constraint is robust to noise miscalibration and

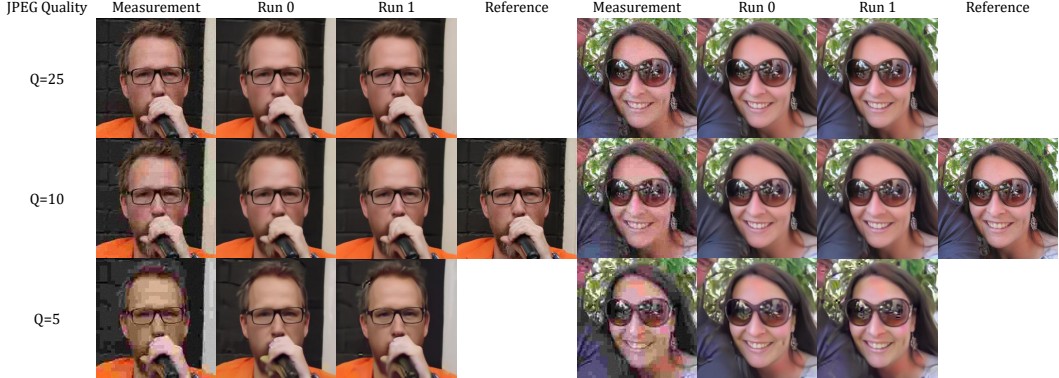

Figure 5: Qualitative results for JPEG restoration on FFHQ using FAST-DIPS with a differentiable surrogate operator. We display the measurement, reconstruction, and the ground-truth reference across three compression levels: JPEG Quality 5, 10, and 25.

Table 8: Quantitative evaluation of JPEG restoration on FFHQ across JPEG quality factors 5, 10, and 25.

| JPEG Quality | PSNR | SSIM | LPIPS |
|---|---|---|---|
| 25 | 31.175 | 0.869 | 0.229 |
| 10 | 29.343 | 0.834 | 0.267 |
| 5 | 26.749 | 0.788 | 0.338 |

Table 9: Quantitative evaluation of high-resolution ($512\times512$) Gaussian deblurring on FFHQ using 10 samples, conducted on an RTX 6000 Ada GPU.

| Method | PSNR | SSIM | LPIPS | Runtime (s) |
|---|---|---|---|---|
| Latent-DAPS | 28.308 | 0.809 | 0.428 | 580.664 |
| **Ours(Latent)** | **31.438** | **0.852** | **0.356** | **247.399** |

remains effective empirically beyond the AWGN setting. Our analytic step-size rules yield stable optimization across tasks, supporting practical insensitivity to corruption type.

**Extension to Non-Differentiable Operators.** To address the applicability of FAST-DIPS to non-differentiable degradations, we evaluated our framework on JPEG restoration. While the standard JPEG compression pipeline involves a non-differentiable quantization step, it can be effectively handled using a differentiable surrogate (Reich et al. (2024)).

We applied FAST-DIPS using this differentiable surrogate to guide the restoration process under measurement noise $\beta = 0.05$. Qualitative results are presented in Figure 5, demonstrating that our method effectively suppresses blocking artifacts and restores high-frequency details. Quantitative metrics in Table 8 further confirm competitive reconstruction performance. These results suggest that FAST-DIPS remains highly effective for formally non-differentiable problems, provided a differentiable proxy of the forward operator is available.

**High-resolution image data.** We further conducted high-resolution experiments on the FFHQ dataset at $512\times512$ resolution in the latent setting, going beyond the standard $256\times256$ regime. As shown in Table 9, our method improves PSNR, SSIM, and LPIPS compared to Latent DAPS—the most recent state-of-the-art latent diffusion–based inverse problem solver—while also achieving approximately $2.3\times$ faster run-time. These results suggest that our hybrid approach can handle high-resolution inputs effectively and benefit from efficient computations in the latent space.

**Qualitative Results.** Figures 6-22 provide additional qualitative samples for a comprehensive set of eight problems on FFHQ and ImageNet datasets. These results visually demonstrate the high-quality and consistent reconstructions achieved by both the pixel-space (FAST-DIPS) and latent-space (Latent FAST-DIPS) versions of our method.

## A.9 FUTURE WORK AND LIMITATIONS

Our proposed method, FAST-DIPS, provides a robust framework for solving inverse problems, and its hyperparameter stability opens up several promising directions for future work. The framework

is defined by a few key hyperparameters ($\rho$, $\varepsilon$, $\sigma_{\text{switch}}$), and as shown in the additional experiments (Tables 6 and 7), it exhibits robustness across a wide range of their values, enhancing its practical usability. Among these, the ADMM penalty parameter $\rho$ can be considered the most influential. While our experiments show stable performance with a fixed value, integrating adaptive penalty selection strategies could further improve convergence and robustness. Similarly, exploring an optimal or adaptive schedule for the hybrid switching point $\sigma_{\text{switch}}$ remains another interesting avenue for research.

Despite these strengths and opportunities, we also acknowledge a primary limitation of the current framework: its dependency on differentiable forward operators. FAST-DIPS is "adjoint-free" in the sense that it does not require a hand-coded adjoint operator. However, its efficiency heavily relies on automatic differentiation to compute VJP and JVP needed for the analytic step size $\alpha^\star$. This implicitly assumes that the forward operator $\mathcal{A}$ is (at least piecewise) differentiable. For problems involving non-differentiable operators or black-box simulators where gradients are unavailable, our current approach cannot be directly applied. Future work could explore extensions using zeroth-order optimization techniques or proximal gradient methods that can handle non-differentiable terms.

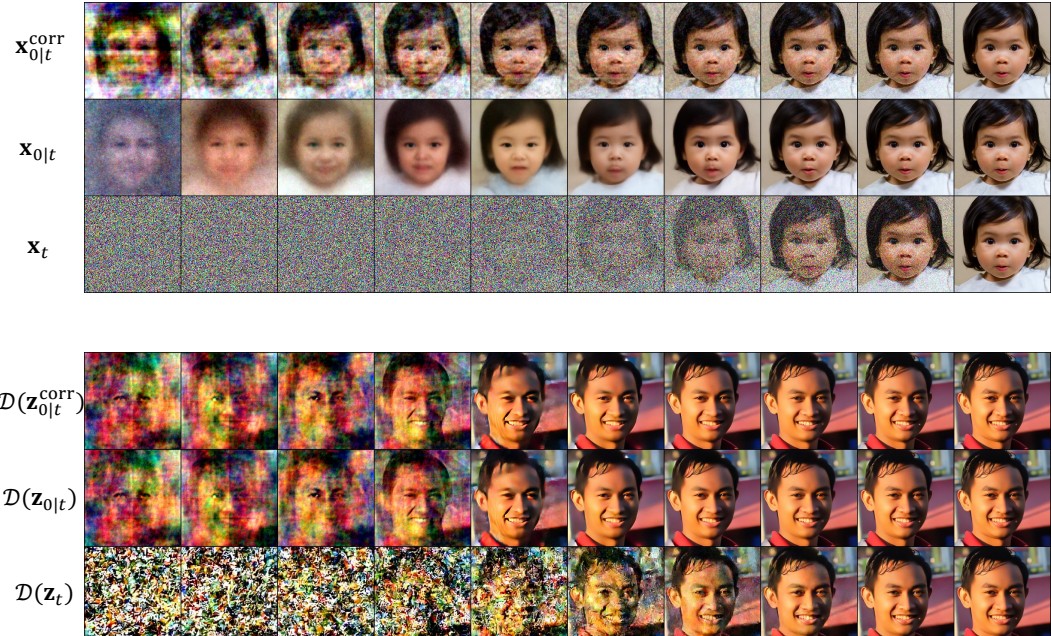

Figure 6: Phase Retrieval trajectory under FAST-DIPS and Latent FAST-DIPS with intermediate iterates along the diffusion schedule.

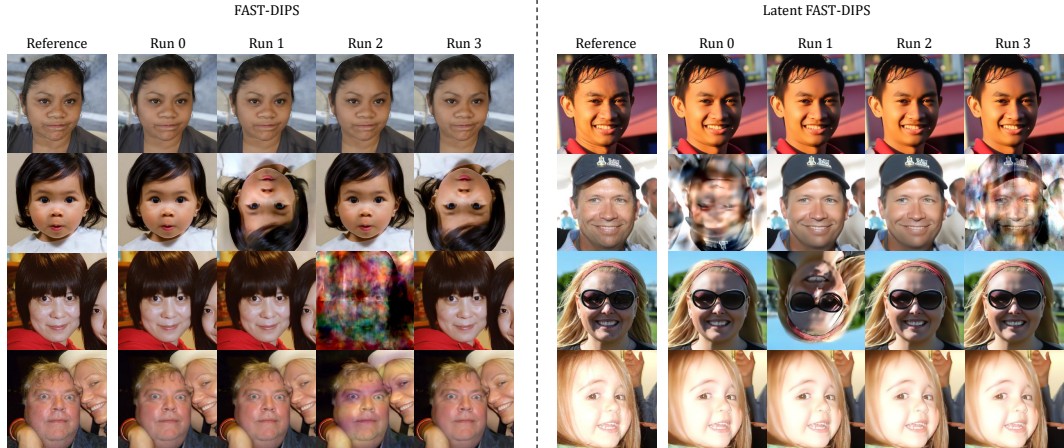

Figure 7: Additional qualitative results for **Phase Retrieval** on FFHQ dataset. We show Reconstruction, and Reference for both FAST-DIPS and Latent FAST-DIPS across four runs (Run 0–3).

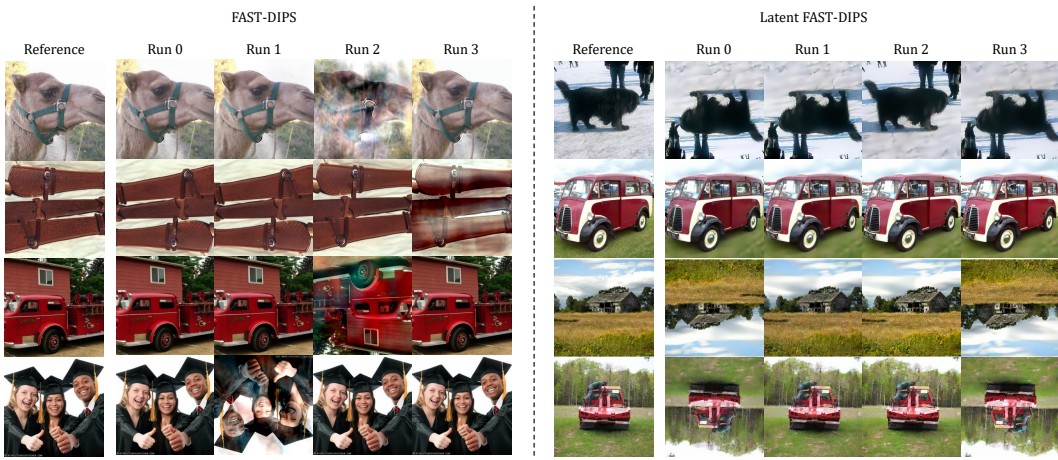

Figure 8: Qualitative results for **Phase Retrieval** on ImageNet dataset. We show Reconstruction, and Reference for both FAST-DIPS and Latent FAST-DIPS across four runs (Run 0–3).

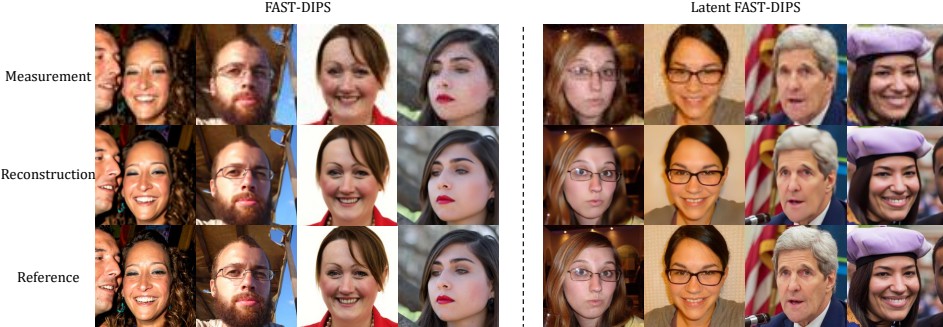

Figure 9: Additional qualitative results for **Super-Resolution** ×4. Measurement, Reconstruction, and Reference are shown for FAST-DIPS and Latent FAST-DIPS.

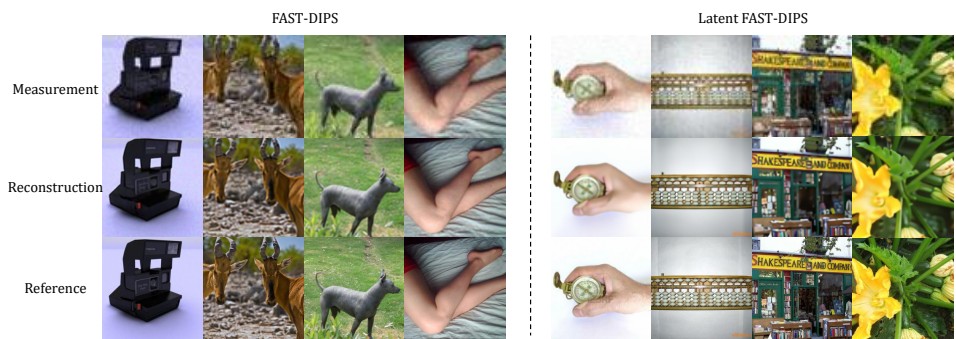

Figure 10: Qualitative results for **Super-Resolution** ×4 on ImageNet dataset. Measurement, Reconstruction, and Reference are shown for FAST-DIPS and Latent FAST-DIPS.

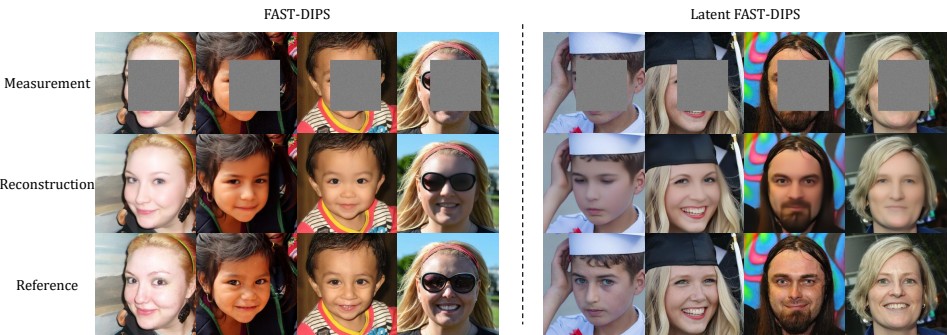

Figure 11: Additional qualitative results for **Inpaint(box)**. We display Measurement, Reconstruction, and Reference for FAST-DIPS and Latent FAST-DIPS.

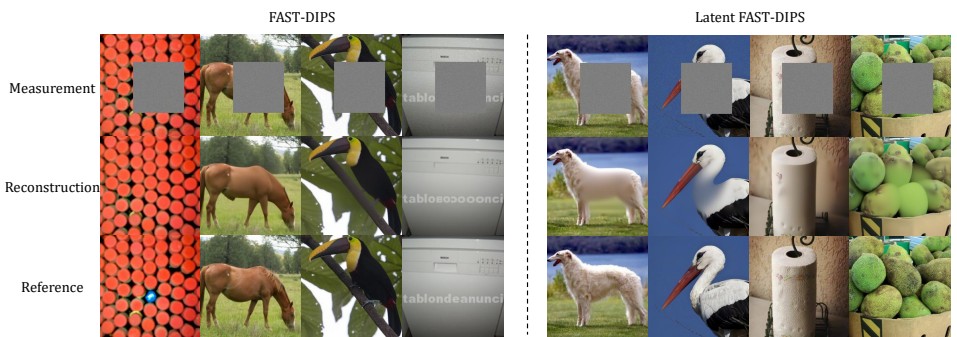

Figure 12: Qualitative results for **Inpaint(box)** on ImageNet dataset. We display Measurement, Reconstruction, and Reference for FAST-DIPS and Latent FAST-DIPS.

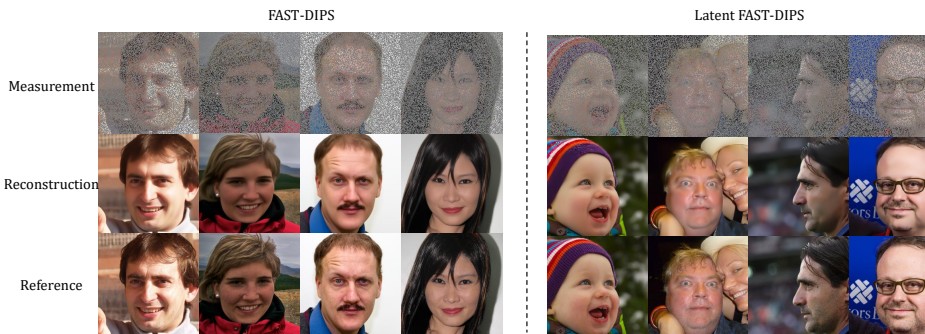

Figure 13: Additional qualitative results for **Inpaint(random)**. Measurement, Reconstruction, and Reference with FAST-DIPS and Latent FAST-DIPS.

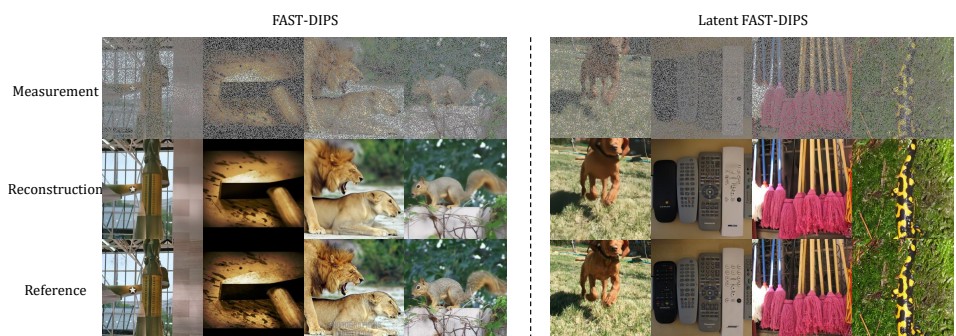

Figure 14: Qualitative results for **Inpaint(random)** on ImageNet dataset. Measurement, Reconstruction, and Reference with FAST-DIPS and Latent FAST-DIPS.

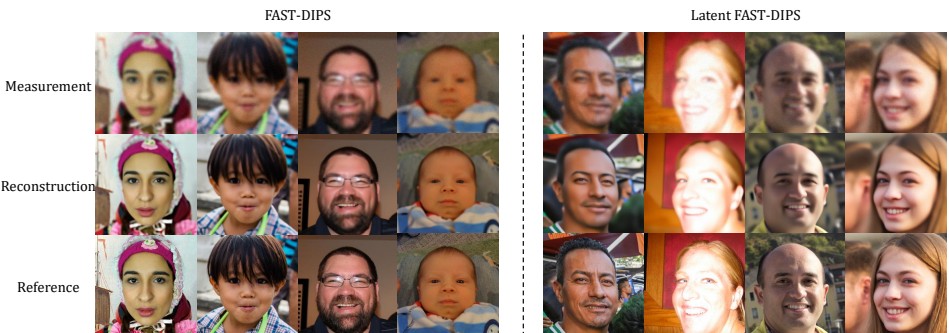

Figure 15: Additional qualitative results for **Gaussian deblurring**. We show Measurement, Reconstruction, and Reference for FAST-DIPS and Latent FAST-DIPS.

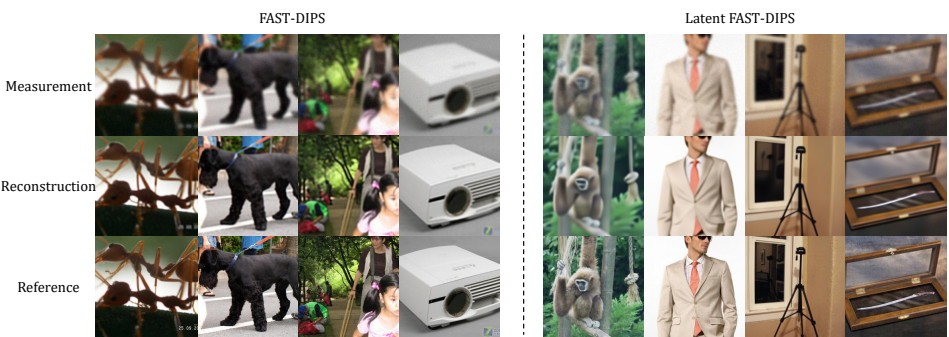

Figure 16: Qualitative results for **Gaussian deblurring** on ImageNet dataset. We show Measurement, Reconstruction, and Reference for FAST-DIPS and Latent FAST-DIPS.

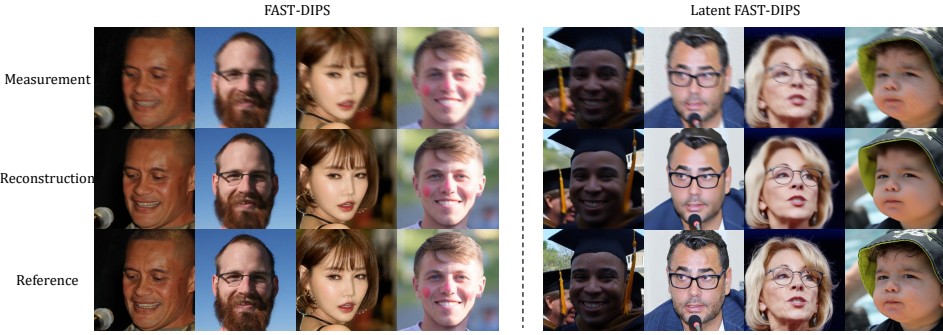

Figure 17: Additional qualitative results for **Motion deblurring**. Measurement, Reconstruction, and Reference are provided for FAST-DIPS and Latent FAST-DIPS.

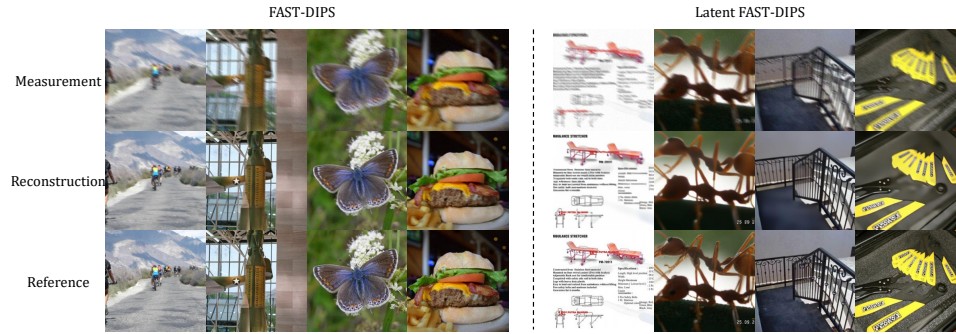

Figure 18: Qualitative results for **Motion deblurring** on ImageNet dataset. Measurement, Reconstruction, and Reference are provided for FAST-DIPS and Latent FAST-DIPS.

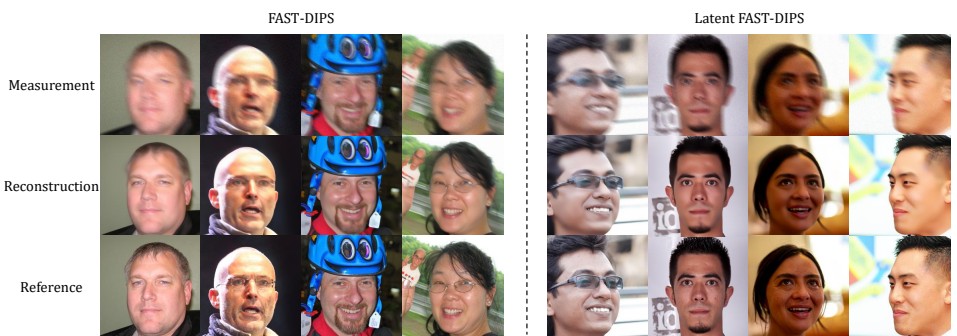

Figure 19: Additional qualitative results for **Nonlinear deblurring**. We present Measurement, Reconstruction, and Reference for FAST-DIPS and Latent FAST-DIPS.

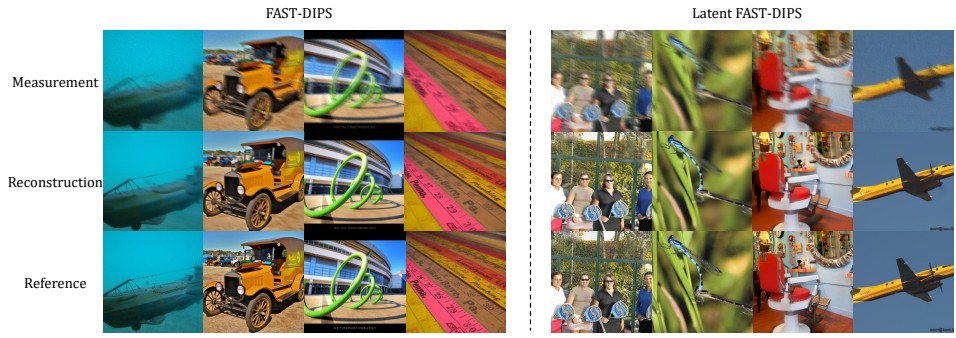

Figure 20: Qualitative results for **Nonlinear deblurring** on ImageNet dataset. We present Measurement, Reconstruction, and Reference for FAST-DIPS and Latent FAST-DIPS.

FAST-DIPS                    Latent FAST-DIPS

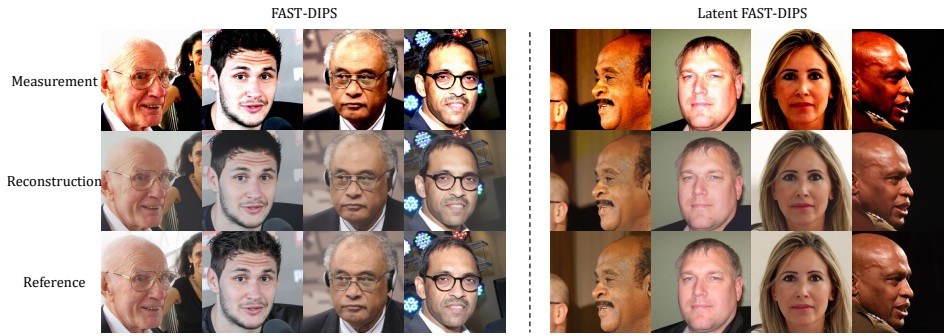

Figure 21: Additional qualitative results for **High Dynamic Range**. Measurement, Reconstruction, and Reference for FAST-DIPS and Latent FAST-DIPS.

FAST-DIPS                    Latent FAST-DIPS

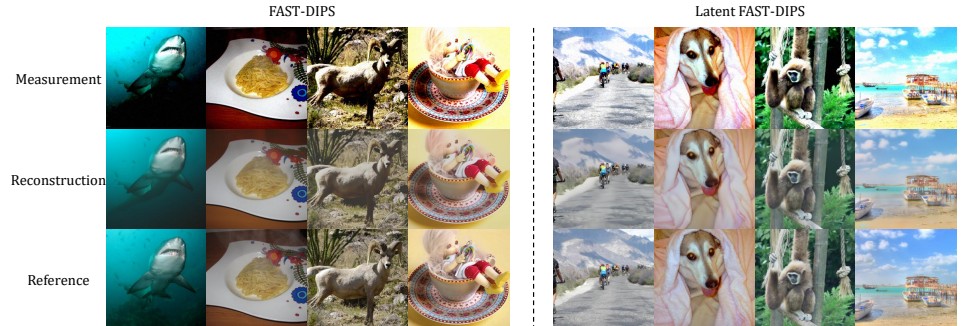

Figure 22: Qualitative results for **High Dynamic Range** on ImageNet dataset. Measurement, Reconstruction, and Reference for FAST-DIPS and Latent FAST-DIPS.

