# OpenReview forum: "FAST‑DIPS: Adjoint‑Free Analytic Steps and Hard‑Constrained Likelihood Correction for Diffusion‑Prior Inverse Problems"
_ICLR.cc/2026/Conference — ICLR 2026 Poster_

### Official Review · Reviewer_9UpX · 2025-10-24

**Soundness:** 3
**Presentation:** 3
**Contribution:** 3
**Rating:** 6
**Confidence:** 3

**Summary:**

This paper introduces FAST-DIPS which aims to improve both the speed and accuracy of existing diffusion-prior solvers. The core contribution is a two-stage update process applied at each step of the reverse diffusion chain: "analytic step" and "hard-constrained likelihood correction" step. The experimental results show that FAST-DIPS consistently outperforms state-of-the-art methods in terms of reconstruction quality and/or computational time.

**Strengths:**

1. The proposed combination of an analytic data consistency step with the hard-constrained likelihood correction is novel and appears highly effective.
2. The paper provides compelling empirical evidence of its superiority, often delivering higher quality in a fraction of the time required by competing methods.
3. The paper is theoretically sound and clearly structured.

**Weaknesses:**

1. The Method section is dense and overly technical, relying heavily on acronyms and mathematical derivations with minimal intuition.

2. The "adjoint-free" claim could be misleading. The method avoids computing the adjoint operator $A^T$ at each iteration by pre-computing a term involving $(A A^T)^{-1}$. This is only advantageous for a specific class of operators where this matrix is easy to compute and invert.

3. The choice of $\epsilon$ in the hard constraint is not well explained

4. The extension of the method to non-linear problems like phase retrieval is handled by linearizing the forward operator at each step. This is a reasonable approach, but the paper provides very little detail or justification for it.

5. The exclusive use of FFHQ restricts the generality of the conclusions.

**Questions:**

1. Could the author please clarify the practical limitations of the adjoint-free formulation? For which classes of forward operators $A$ does the pre-computation of $(A A^T)^{-1} A$ become a bottleneck that outweighs the per-iteration speed-up?

2. Though the experiment section has included various inverse problems, the selection of the dataset is very limited. Can the author please add at least one more dataset other than human face to benchmark the performance?

3, The hybrid schedule introduces a switching threshold $\sigma_\text{switch}$. How is this parameter selected in practice, and how does performance vary with its value?

---

> ### Author Response · Authors · 2025-11-22
> **Author response (1/2)**
>
> Thank you for the detailed review and constructive feedback. The following is our response to the specific comments and questions.
>
> > W2. The ``adjoint-free'' claim could be misleading. The method avoids computing the adjoint operator $A^T$ at each iteration by pre-computing a term involving $(AA^T)^{-1}$. This is only advantageous for a specific class of operators where this matrix is easy to compute and invert.
>
> > Q1. Could the author please clarify the practical limitations of the adjoint-free formulation? For which classes of forward operators $A$ does the pre-computation of $(AA^T)^{-1}A$ become a bottleneck that outweighs the per-iteration speed-up?
>
> We clarify that FAST-DIPS neither forms nor inverts $(AA^\top)$, nor do we pre-compute $(AA^\top)^{-1}A$. Our implementation does not use normal equations or explicit matrix inverses. When we say adjoint-free, we mean that the user only needs to implement the forward map $\mathcal{A}(\textbf{x})$; all required Jacobian–vector and vector–Jacobian products are obtained via standard automatic differentiation (or a single finite-difference probe), for both linear and nonlinear operators.
>
> > W5. The exclusive use of FFHQ restricts the generality of the conclusions.
>
> > Q2. Though the experiment section has included various inverse problems, the selection of the dataset is very limited. Can the author please add at least one more dataset other than human face to benchmark the performance?
>
> We agree that evaluating solely on FFHQ limits the assessment of the method's generality. To address this, we have conducted additional experiments on the ImageNet ($256 \times 256$) dataset. We have revised the manuscript to include these new quantitative comparisons in main paper Table 1 and qualitative samples in Figures 7-22. These results demonstrate that FAST-DIPS generalizes effectively to the diverse structural and textural content of natural images, maintaining its performance and speed advantages beyond the domain of human faces.
>
> > W1. The Method section is dense and overly technical, relying heavily on acronyms and mathematical derivations with minimal intuition.
>
> We agree that, in the original submission, the Method section jumps quickly into notation and derivations, which can make the presentation feel dense.
>
> To improve intuition, we have added a new subsection ``High-level overview and design choices'' (Section 2.1) at the beginning of the Method section. This subsection explains in plain language:
> (i) what practical issues FAST-DIPS aims to address,
> (ii) how the method is structured at a high level, and
> (iii) which theoretical results support each design choice.
>
> Immediately following this, we added a short ``Practical per-step recipe'' paragraph that describes one reverse diffusion step in three stages, and spells out what is evaluated at each stage. Readers can therefore first grasp the overall logic and computational structure of FAST-DIPS from these two paragraphs, and only then dive into the detailed mathematical derivations if desired.
>
> We hope this front-loaded overview mitigates the perceived density of the Method section and provides the missing intuition.

---

> ### Author Response · Authors · 2025-11-22
> **Author response (2/2)**
>
> > W3. The choice of $\epsilon$ in the hard constraint is not well explained
>
> In all eight inverse problems we use a unified and simple rule for setting $\epsilon$. After normalizing images to a fixed dynamic range, we interpret $\epsilon$ as a tolerance on the normalized data residual $\\|\mathcal{A}(\mathbf{x})-\mathbf{y}\\|$. For each operator family (blur/SR, inpainting, phase retrieval, nonlinear blur, HDR), we run a small validation sweep over a few candidates (e.g. $\epsilon \in \{0, 0.01, 0.1, 0.5, 1.0\}$) on a held-out subset. As reported in Appendix Table 6, the reconstruction metrics remain essentially unchanged over more than two orders of magnitude in $\epsilon$ once it is small but nonzero. Guided by this robustness, we fix a single conservative value, $\epsilon = 0.05$ in normalized units, for each operator family and reuse it across all experiments, without per-instance tuning.
>
> > W4. The extension of the method to non-linear problems like phase retrieval is handled by linearizing the forward operator at each step. This is a reasonable approach, but the paper provides very little detail or justification for it.
>
> The key point is how the relation
> $
> \mathcal{A}(\mathbf{x} - \alpha \mathbf{g})
> \approx
> \mathcal{A}(\mathbf{x}) - \alpha\ J_\mathcal{A}(\mathbf{x})\ \mathbf{g}
> $ behaves in the linear vs. non-linear cases. If $\mathcal{A}$ is linear, then
> $J_\mathcal{A}(\mathbf{x}) \equiv \mathcal{A}$ and the expression above holds
> exactly; the resulting quadratic model of $F(\mathbf{x} - \alpha \mathbf{g})$
> along $-\mathbf{g}$ is exact, and the closed-form $\alpha^\star$ we derive is
> the true optimal steepest-descent step for $F$ in that direction. For
> non-linear $\mathcal{A}$, the same expression is the first-order Taylor
> expansion with a higher-order remainder term. Proposition 3 shows that under
> mild smoothness this remainder only induces a higher-order error in the model
> (of order $O(\||\mathbf{g}\||^3)$), and that the step obtained from the quadratic
> approximation, together with backtracking, still yields a genuine descent step
> for the true non-linear objective $F$. In the revised manuscript we now
> explicitly state this distinction (exact in the linear case, first-order with a
> controlled residual in the non-linear case) immediately after introducing the
> linearization, and justify the approximation by pointing to the descent
> guarantee provided by Proposition 3.
>
>
> > Q3. The hybrid schedule introduces a switching threshold $\sigma_{\text{switch}}$. How is this parameter selected in practice, and how does performance vary with its value?
>
> Regarding the sensitivity and selection of $\sigma_{\text{switch}}$, we illustrated the performance trade-offs in Table 7 of the Appendix. In our experiments, we adopted a task-dependent strategy: we set $\sigma_{\text{switch}}=1$ for linear tasks and $\sigma_{\text{switch}}=5$ for nonlinear tasks. Our rationale for this selection is that nonlinear inversion is inherently more difficult and prone to instability. By choosing a larger $\sigma_{\text{switch}}$ for nonlinear tasks, the method transitions from pixel-space to latent-space corrections earlier in the diffusion schedule. This earlier switch is designed to minimize error accumulation from encoder-decoder discrepancies and to increase the weight of the more precise $z$-space updates, which better leverage the generative manifold to regularize the complex inversion process.
>
> We hope our response adequately addresses your concerns. We are happy to provide further clarifications if needed.

---

### Official Review · Reviewer_pSwc · 2025-10-26

**Soundness:** 3
**Presentation:** 3
**Contribution:** 3
**Rating:** 8
**Confidence:** 4

**Summary:**

This paper proposes FAST-DIPS, a fast and training-free solver for diffusion-prior inverse problems. The key idea is to perform a hard-constrained likelihood correction at each diffusion step through an adjoint-free ADMM scheme with analytic step size computation, thereby avoiding the need for hand-coded adjoints or inner MCMC loops.
Experimental results across multiple linear and nonlinear inverse problems demonstrate that FAST-DIPS achieves comparable or superior reconstruction quality while reducing runtime compared to state-of-the-art training-free baselines.

**Strengths:**

1. **Clear and principled framework**:
The paper presents FAST-DIPS, a training-free and adjoint-free framework for diffusion-prior inverse problems, offering a clean and principled alternative to existing plug-and-play or posterior-sampling approaches.

2. **Broad applicability and ease of use**:
The method supports both pixel-space and latent-space diffusion models through an adjoint-free design, making it versatile and easy to apply to a variety of inverse problems without hand-crafted adjoints or retraining.

2. **Strong empirical performance**:
The proposed analytic-step ADMM correction eliminates the need for hand-crafted adjoints or inner MCMC loops, making the method broadly applicable to both linear and nonlinear operators while achieving 5×–25× faster inference.

**Weaknesses:**

1. **Presentation issue**:
Table 1 exceeds the page width and is difficult to read in its current format. The authors should consider reformatting or splitting the table across pages to improve readability and compliance with ICLR formatting guidelines.

**Questions:**

1.The performance of the baseline algorithms in Table 1 differs from that reported in their original papers, even under the same experimental settings. For instance, the SITCOM paper reports a PSNR of 30.68 for the SR task, whereas Table 1 reports 29.555. A similar discrepancy is also observed for the DAPS algorithm. While Figure 3 effectively demonstrates the superiority of the proposed method under the same runtime, I would appreciate clarification regarding these inconsistencies.

---

> ### Author Response · Authors · 2025-11-22
> **Author response**
>
> Thank you for the detailed review and constructive feedback. The following is our response to the specific comments and questions.
>
> > W1. Presentation issue: Table 1 exceeds the page width and is difficult to read in its current format. The authors should consider reformatting or splitting the table across pages to improve readability and compliance with ICLR formatting guidelines.
>
> We have extensively revised and reformatted the table to strictly adhere to the ICLR formatting guidelines and fit within the page margins. This reorganization was also necessary to accommodate the new quantitative results from our additional ImageNet experiments, ensuring the data is presented clearly and comprehensively in the revised manuscript.
>
> > Q1. The performance of the baseline algorithms in Table 1 differs from that reported in their original papers, even under the same experimental settings. For instance, the SITCOM paper reports a PSNR of 30.68 for the SR task, whereas Table 1 reports 29.555. A similar discrepancy is also observed for the DAPS algorithm. While Figure 3 effectively demonstrates the superiority of the proposed method under the same runtime, I would appreciate clarification regarding these inconsistencies.
>
> For the FFHQ dataset, all of our reported numbers (ours and baselines) are computed on the same validation split: specifically, we evaluate on 100 images with indices 00000–00099.
>
> In contrast, for DAPS, the authors provide a validation list on their project page with indices 49000–49999 (1000 images), and the paper states that they select 100 images from this pool, but does not specify which subset is used. Similarly, the SITCOM paper reports using 100 FFHQ validation images, but the exact indices are not disclosed. As a result, it is not possible for us to exactly reproduce the original validation split for these methods, and we expect this mismatch in the evaluation subset to account for the small discrepancies in Table 1 and the numbers reported in the original papers.
>
> Importantly, all results in Table 1 are obtained by running the baseline implementations and our method on the same FFHQ validation split and under the same experimental settings.
>
> We hope our response adequately addresses your concerns. We are happy to provide further clarifications if needed.

---

> > ### Comment · Reviewer_pSwc · 2025-11-24
> >
> > I thank the authors for their response to my review. I am satisfied to keep my score where it is.

---

### Official Review · Reviewer_6kKx · 2025-11-01

**Soundness:** 3
**Presentation:** 2
**Contribution:** 2
**Rating:** 4
**Confidence:** 5

**Summary:**

This paper introduces FAST-DIPS, a training-free diffusion-prior inverse problem solver. The core contributions are: (1) an adjoint-free correction step that (2) enforces a hard-constrained likelihood ($||\mathcal{A}(x)-y||\le\epsilon$) using (3) an ADMM formulation, where the primal update is solved efficiently with an analytic, non-iterative step size derived from a local quadratic model. The method is further extended to latent and hybrid (pixel/latent) settings.

**Strengths:**

1. The adjoint-free ADMM formulation with analytic step sizes (via VJP/JVP or finite-difference approximations) is a clever way to minimize engineering overhead while ensuring efficient and deterministic updates.

2. The paper provides strong local guarantees, including exact minimization of quadratic models (Proposition 3), KKT satisfaction at fixed points (Proposition 4), and descent properties with backtracking.

**Weaknesses:**

1. My main concern is that, while adjoint-free, the method still requires autodiff through $\mathcal{A}$, and the latent mode incurs repeated decoder calls. In my view, the engineering benefit of avoiding adjoints is somewhat offset by the computational cost of automatic differentiation through $\mathcal{A}$.

2. The method’s hard constraint ($||\mathcal{A}(x)-y||\le\epsilon$) shifts the tuning burden from the likelihood weight to the credible set’s radius $\epsilon$. Although this is briefly acknowledged in the limitations section, the paper should more thoroughly discuss how this critical hyperparameter is selected.

3. Experiments are limited to FFHQ (faces only), which lacks diversity; no tests are conducted on broader datasets such as CelebA-HQ, LSUN, or natural images (e.g., ImageNet subsets). This raises concerns about generalization to non-face domains or higher resolutions.

4. The complete algorithm, as presented in the appendix, is somewhat complex, making the method less elegant. Moreover, the writing could be improved for better clarity and flow.

**Questions:**

1. How was the hyperparameter $\epsilon$ (the radius of the credible set) determined for each of the eight experiments?

2. The paper claims to use no inner MCMC, but the ADMM iterations ($K=3$–$5$) with $S=1$ descent appear to form mini-loops. Please clarify if I have misunderstood this point.

3. For the latent variant, have you analyzed the computational cost of the JVP $J_{\mathcal{A}\circ\mathcal{D}}(z)g$?

---

> ### Author Response · Authors · 2025-11-22
> **Author response (1/3)**
>
> Thank you for the detailed review and constructive feedback. The following is our response to the specific comments and questions.
>
> > W1. My main concern is that, while adjoint-free, the method still requires autodiff through $\mathcal{A}$, and the latent mode incurs repeated decoder calls. In my view, the engineering benefit of avoiding adjoints is somewhat offset by the computational cost of automatic differentiation through $\mathcal{A}$.
>
> |Method| Res.|Total (s)|Latent corr. (s)|Forward (s)|VJP (s)|JVP (s)|Peak VRAM (GiB)|
> |-|-|-|-|-|-|-|-|
> | Latent DAPS| $256^2$ |84.97|-- |--|--|--|4.40|
> | Latent DAPS| $512^2$ |560.80|--|--|--|--|11.28|
> | Latent-FAST-DIPS (ours)|$256^2$ |38.38| 36.73|13.52| 5.22| 17.31|4.41|
> | Latent-FAST-DIPS (ours)|$512^2$ |243.61| 237.48|90.25|37.04|108.42|11.32|
>
> *Table 1* Scalability of latent solvers on FFHQ at $256^2$ and $512^2$ (10 samples, batch size 1).
>
> We agree that avoiding manual adjoint derivation entails running automatic differentiation (VJP and JVP) through $\mathcal{A}$ and, in the latent case, the decoder. However, our analysis shows that this trade-off remains favorable for two reasons: generality and efficiency.
>
> **Generality and Applicability.**
> The primary advantage of being adjoint-free is not merely avoiding the implementation effort of deriving $\mathcal{A}^T$, but extending applicability to non-linear or complex operators where an analytical adjoint is difficult or impossible to derive (e.g., high-dynamic-range tone mapping, neural network-based forward models, or phase retrieval). By relying on standard VJP/JVP interfaces—which are intrinsic to modern deep learning frameworks—our framework allows users to plug in any differentiable $\mathcal{A}$ immediately.
>
> **Net Computational Efficiency.**
> While autodiff introduces a per-iteration overhead, it enables the calculation of an optimal step size (via the JVP), which drastically reduces the total number of iterations required for convergence.
> As shown in Table 1, although the per-step cost is higher than methods like DAPS (which rely on cheaper, heuristic steps), the total runtime is reduced by a factor of $\approx 2.3\times$.
> Furthermore, regarding the cost of autodiff through $\mathcal{A}$: modern reverse-mode autodiff (VJP) generally has the same asymptotic time complexity as a manually implemented adjoint operator (typically within a constant factor of $2\text{--}3\times$ the forward pass). Therefore, the ``penalty'' of using autodiff is bounded and, as our results confirm, is effectively amortized by the reduction in sampling steps.
>
> Thus, the engineering benefit is not offset by the cost; rather, the cost is an investment that pays off in both wider applicability (handling black-box/non-linear $\mathcal{A}$) and faster overall convergence.
>
> > W2. The method’s hard constraint ($\\| \mathcal{A}(x)-y\\| \leq \epsilon$) shifts the tuning burden from the likelihood weight to the credible set’s radius $\epsilon$. Although this is briefly acknowledged in the limitations section, the paper should more thoroughly discuss how this critical hyperparameter is selected.
>
> We acknowledge that the hard constraint $\\|\mathcal{A}(x)-y\\|\leq\epsilon$ introduces $\epsilon$ as a parameter. However, unlike the likelihood weight in standard energy-based methods---which globally alters the gradient landscape and requires careful balancing against the prior score---$\epsilon$ has a clear geometric interpretation: it defines the radius of the feasible set (the ``noise ball'') within which the solution must lie.
>
> **Selection Strategy and Interpretation.**
> In many practical inverse problems (e.g., blind restoration), the exact measurement noise level is unknown. In such cases, our strategy is to set $\epsilon$ to a small constant (e.g., $\epsilon=0.05$) to enforce tight data consistency.
> This choice acts as a numerical relaxation of the exact projection constraint ($\\|\mathcal{A}(x)-y\\|=0$). By setting $\epsilon$ to a small non-zero value rather than zero, we ensure high fidelity to the measurement while allowing a minute margin for numerical errors inherent in iterative variable-splitting solvers. This prevents the brittleness associated with strict equality constraints while effectively treating the measurement as a strong anchor.
>
> **Robustness Analysis.**
> We find that the method is insensitive to the exact value of $\epsilon$ provided it remains small. As shown in Table 6 in Appendix, varying $\epsilon$ across orders of magnitude results in negligible performance differences.
>
> For $\epsilon \in [0, 1]$, the PSNR and LPIPS are effectively identical. This flatness in the sensitivity curve indicates that the ``tuning burden'' is minimal in practice: the generative prior is robust enough to reconstruct high-quality details as long as the solution is constrained to the local measurement subspace. To ensure full reproducibility, we will explicitly specify how we set this value ($\epsilon=0.05$) in Table 2 of the revised Appendix.

---

> ### Author Response · Authors · 2025-11-22
> **Author response (2/3)**
>
> > W3. Experiments are limited to FFHQ (faces only), which lacks diversity; no tests are conducted on broader datasets such as CelebA-HQ, LSUN, or natural images (e.g., ImageNet subsets). This raises concerns about generalization to non-face domains or higher resolutions.
>
> |**Method**|**PSNR**|**SSIM**|**LPIPS**|**Runtime (s)**|
> |-|-|-|-|-|
> |Latent-DAPS|28.308| 0.809|0.428|580.664|
> |**Latent-FAST-DIPS (ours)**|**31.438**|**0.852**|**0.356**|**247.399**|
>
> *Table 2* High-resolution ($512\times512$) Gaussian deblurring results on FFHQ using latent diffusion priors.
>
> We agree that the original FFHQ-only evaluation did not fully demonstrate generalization beyond faces and $256^2$ resolution. In the revised manuscript we therefore added two sets of experiments.
>
> **Natural-image generalization.** We now evaluate all eight inverse problems on a diverse $256\times256$ ImageNet subset using pre-trained diffusion model (Dhariwal \& Nichol [1]). Table 1 in the revised paper reports FFHQ and ImageNet results side by side: across both domains FAST-DIPS consistently matches or improves PSNR/SSIM/LPIPS over training-free baselines while retaining the same runtime advantage. This shows that the method does not rely on facial structure and transfers to generic natural images.
>
> **High-resolution latent setting.** We also have performed a $512\times512$ Gaussian deblurring experiment on FFHQ in the latent setting as shown in Table 2 in this comment. Latent FAST-DIPS improves PSNR from 28.308 to 31.438, SSIM from 0.809 to 0.852, LPIPS from 0.428 to 0.356, and reduces runtime from 580.7s to 247.4s ($\approx 2.3\times$ speedup) compared to Latent-DAPS. These new ImageNet and $512^2$ results, now included in the paper, demonstrate that FAST-DIPS generalizes beyond faces and remains efficient at higher resolutions.
>
> [1] Prafulla Dhariwal and Alex Nichol. Diffusion Models Beat GANs on Image Synthesis. NeurIPS 2021.
>
> > W4. The complete algorithm, as presented in the appendix, is somewhat complex, making the method less elegant. Moreover, the writing could be improved for better clarity and flow.
>
> In the revision we have explicitly separated concept from implementation to improve elegance and flow:
> * We added a new subsection ``High-level overview and design choices'' (Section 2.1) that explains, in plain language, what problems FAST-DIPS aims to solve (costly inner gradient/MCMC loops, sensitive likelihood weights), why we adopt local hard-constrained MAP + adjoint-free ADMM + re-annealing, and how these choices connect to the theoretical guarantees (KL control, descent, KKT feasibility).
>  * To mitigate the “complex-looking” full algorithm, we added a short “Practical per-step recipe” paragraph that summarizes one reverse diffusion step in three stages (denoiser proposal $\to$ $K$ fast ADMM updates $\to$ re-annealing), including what is evaluated per iteration (one forward, one VJP, one JVP). This would play the role of a minimal algorithm description.
>
> We believe these changes improve the clarity and perceived elegance of the method, while preserving all the technical detail needed for reproducibility.

---

> ### Author Response · Authors · 2025-11-22
> **Author response (3/3)**
>
> > Q1. How was the hyperparameter $\epsilon$ (the radius of the credible set) determined for each of the eight experiments?
>
> For all eight inverse problems we follow the same simple procedure. After normalizing images to a fixed dynamic range, we regard $\epsilon$ as a tolerance on the normalized data residual and run a small validation sweep over a few candidate values (e.g. $\epsilon \in \{0, 0.01, 0.05, 0.1, 1.0\}$) on a held-out subset for each operator family. As shown in the sensitivity study in Appendix Table 6, reconstruction metrics are essentially flat over more than two orders of magnitude in $\epsilon$ once it is “small-but-nonzero”. Based on this, we select a single small value (e.g. $\epsilon = 0.05$ in normalized units) per operator family and reuse it for
> all experiments.
>
> In the revised manuscript we will make this explicit by listing, in Appendix Table 2, the exact $\epsilon$ used for each of the eight tasks.
>
> > Q2. The paper claims to use no inner MCMC, but the ADMM iterations ($K=3-5$) with $S=1$ descent appear to form mini-loops. Please clarify if I have misunderstood this point.
>
>
> Our claim of using ``no inner MCMC'' is meant in the sense that FAST-DIPS does not run an additional stochastic Markov chain (e.g., Langevin or Metropolis steps) inside each diffusion step. Instead, at each noise level we apply a small, fixed number of deterministic ADMM updates.
>
> Thus, while there is indeed a short inner optimization loop, it is not an inner MCMC chain:
> * The ADMM iterations perform deterministic proximal updates that monotonically decrease the augmented Lagrangian, and introduce no additional randomness beyond the outer diffusion noise.
> * In contrast, methods such as DAPS/SITCOM run an inner stochastic chain per diffusion step (e.g., tens to hundreds of Langevin/annealing steps), where each iteration injects noise and explores a Markov trajectory in state space.
>
> Our intention with ``no inner MCMC'' is to highlight that FAST-DIPS replaces these long stochastic inner chains with a short, deterministic refinement loop whose depth is kept small and fixed across experiments.
>
> > Q3. For the latent variant, have you analyzed the computational cost of the JVP $J_{\mathcal{A} \circ\mathcal{D}}(z)g$
>
> We explicitly analyzed the cost of the JVP term $J_{\mathcal{A} \circ\mathcal{D}}(z)g$ and found that while it is computationally heavier per call, it significantly reduces the total runtime by minimizing the number of required function evaluations (NFE).
>
> **Cost per step vs. Total runtime.**
> As shown in Table 1 in previous comment, the JVP is indeed the most expensive operation in our correction step (taking $\approx 108$s at $512^2$, or roughly $46\\%$ of the correction time). This is because the JVP (forward-mode AD) is approximately $3\times$ slower than the VJP (reverse-mode AD) in our PyTorch implementation. However, the overall computational cost is determined by the product of the cost per step and the total number of steps needed to converge.
>
> Standard methods like DAPS typically rely on fixed or heuristic step sizes (e.g., Langevin dynamics), which are cheap to compute per iteration but require a large number of steps to satisfy data consistency. In contrast, FAST-DIPS uses the JVP to compute an optimal step size analytically at each iteration. This allows us to take large, accurate steps that satisfy data consistency with significantly fewer evaluations.
>
> **Empirical Validation.**
> This trade-off is validated by our runtime results. Although our per-step cost is higher due to the JVP, Latent FAST-DIPS is $\approx 2.3\times$ faster than Latent DAPS overall ($243.61$s vs. $560.80$s at $512^2$). The computational overhead of the JVP is effectively amortized by the drastic reduction in the total number of iterations required to solve the inverse problem.
>
> We hope our response adequately addresses your concerns. We are happy to provide further clarifications if needed.

---

### Official Review · Reviewer_mDLA · 2025-11-04

**Soundness:** 2
**Presentation:** 3
**Contribution:** 3
**Rating:** 6
**Confidence:** 2

**Summary:**

This paper introduces FAST-DIPS, a training-free solver for diffusion-prior inverse problems, including those with nonlinear forward operators. The method's core is a hard-constrained proximal correction via an adjoint-free ADMM. This approach replaces costly inner MCMC loops or iterative optimization with an analytic step size, computable from one VJP and one JVP. Experiments across eight linear and nonlinear tasks demonstrate comparable or superior quality to state-of-the-art baselines, while achieving faster runtimes.

**Strengths:**

1.  The method’s adjoint-free design is both interesting and practical, eliminating the need for hand-coded adjoints. By relying on standard automatic differentiation (VJP/JVP), the framework is directly applicable to a broad class of nonlinear inverse problems—a setting that remains challenging for many competing methods.

2.  The framework is evaluated extensively and demonstrates a substantial speedup over baselines such as DAPS by replacing costly inner MCMC loops with an analytic step size, while maintaining—often even improving—reconstruction quality.

**Weaknesses:**

1.  One concern is the method's reliance on differentiable forward operators. The entire framework is built upon the availability of VJP and JVP, making it inapplicable to common non-differentiable degradations such as JPEG restoration or quantization. This may limit its utility for many real-world degradation types.

2.  Another concern is the need for task-specific hyperparameter tuning. The authors show that key parameters were set to different values for each of the eight tasks. This implies that users must perform a new, potentially expensive hyperparameter search for any novel problem, undermining its practicality as a plug-and-play solver. Moreover, I wonder if this hyperparameter selection is essential for other baselines as well. If so, we need to include the cost of hyperparameter selection to show practical speed-up in the usage of the proposed method.

3. Moreover, I am skeptical about the method's scalability to higher resolutions. The computational cost of the VJP/JVP, especially for the latent variant, which requires backpropagation through the decoder, was manageable for the $256 \times 256$ experiments. However, this computation and memory overhead is likely to become a significant bottleneck as resolution increases, potentially limiting the framework's applicability to large-scale problems.

**Questions:**

Please see the Weaknesses for the details.

---

> ### Author Response · Authors · 2025-11-22
> **Author response (1/3)**
>
> Thank you for the detailed review and constructive feedback. The following is our response to the specific comments and questions.
>
> > W1. One concern is the method's reliance on differentiable forward operators. The entire framework is built upon the availability of VJP and JVP, making it inapplicable to common non-differentiable degradations such as JPEG restoration or quantization. This may limit its utility for many real-world degradation types.
>
> |JPEG Quality|PSNR|SSIM|LPIPS|
> |-|-|-|-|
> |25|31.175|0.869|0.229|
> |10|29.343|0.834|0.267|
> |5|26.749|0.788|0.338|
>
> *Table 1* Quantitative evaluation of JPEG restoration on FFHQ across JPEG quality factors 5, 10, and 25 under measurement noise $\beta = 0.05$.
>
> We agree that our theoretical analysis assumes a differentiable forward operator $\mathcal{A}$; this is stated explicitly in our assumptions and is shared by most gradient-based diffusion-prior solvers (e.g., DPS-style methods, plug-and-play with diffusion priors), which all require access to $\nabla_\textbf{x} \\|\mathcal{A}(\textbf{x}) - \textbf{y}\\|$ or related quantities.
>
> While many real-world degradation operators are inherently non-differentiable, established methods exist to surrogate a non-differentiable operator with a differentiable one. For instance, specific non-differentiable components can be approximated by smooth functions:
> * Hard rounding can be replaced by a smooth ``soft-round'' map (e.g., using sigmoids/softmax around neighboring bins).
> * Clipping can be replaced by a smooth saturating nonlinearity.
> * Blockwise quantization can be modeled as a differentiable block transform followed by additive noise.
>
> Through such approaches, a differentiable operator can approximate the real-world non-differentiable process.
>
> As a concrete example of this technique, we conducted an experiment on JPEG. We utilized the differentiable JPEG of Reich et al. [1] to apply the forward operator within our framework. This is crucial, as the actual measurements $y$ were still generated using the standard, non-differentiable `torchvision.io` `encode_jpeg` and `decode_jpeg` functions. The visual results demonstrating the effectiveness of this approximation were included in the figures of the revised paper.
>
> Furthermore, we anticipate that our method can be extended to more realistic scenarios involving multiple or unknown degradations. Man et al. [2] propose predicting the parameters of each degradation operator from the measurements. We expect that combining such operator-parameter prediction with our framework could enable our approach to handle measurements corrupted by unknown real-world degradations.
>
> [1] Christoph Reich, Biplob Debnath, Deep Patel, and Srimat Chakradhar. Differentiable JPEG: The Devil is in the Details. WACV 2024.
>
> [2] Sean Man, Guy Ohayon, Ron Raphaeli, and Michael Elad. Proxies for Distortion and Consistency with Applications for Real-World Image Restoration. arXiv:2501.12102, 2025.

---

> ### Author Response · Authors · 2025-11-22
> **Author response (2/3)**
>
> > W2. Another concern is the need for task-specific hyperparameter tuning. The authors show that key parameters were set to different values for each of the eight tasks. This implies that users must perform a new, potentially expensive hyperparameter search for any novel problem, undermining its practicality as a plug-and-play solver. Moreover, I wonder if this hyperparameter selection is essential for other baselines as well. If so, we need to include the cost of hyperparameter selection to show practical speed-up in the usage of the proposed method.
>
> We agree that a practical solver should not require heavy, problem-specific tuning. In the revised paper we have simplified the parameterization so that the only “true” method-specific hyperparameters are the ADMM penalty $\rho$ and the hard-constraint radius  $\varepsilon$; the diffusion trust-region scale is fixed via $\gamma_t = \sigma_t^2$ (we removed $\lambda$ since it can be absorbed into
> $\rho$). All other parameters ($T,K,S,\sigma_{\text{switch}}$) mainly control
> how much computation one is willing to spend and can be adjusted according to
> a given compute budget.
>
> **Robustness of $\rho$ and $\varepsilon$.**
> Appendix Table 6 shows that FAST-DIPS is quite insensitive to these two
> hyperparameters. For $\varepsilon$, sweeping across more than two orders of
> magnitude (from $0$ to $1.0$) produces essentially identical reconstruction
> quality; PSNR and LPIPS change only in the third decimal place once
> $\varepsilon$ is small:
> $
> \varepsilon \in \{0,0.01,0.05,0.1,1.0\}
> \ \Rightarrow\
> \text{PSNR} \approx 29.73\text{ dB},\ \text{SSIM} \approx 0.839,\ \text{LPIPS} \approx 0.255\text{–}0.258.
> $
> Based on this flat sensitivity, we simply fix $\varepsilon = 0.05$ for all tasks and for both pixel and latent variants, without any task-specific tuning. For $\rho$, Table 6 shows that performance is also stable over a wide range ($10$ to $1000$): PSNR is already strong at $\rho = 100$, peaks around $200$, and degrades only mildly outside this range. In practice, a coarse sweep over a small grid (e.g., $\rho \in \{5, 200\}$) is sufficient to find a good setting; there is no need for an expensive, fine-grained search.
>
> **Other parameters as compute knobs.**
> The remaining parameters ($T,K,S,\sigma_{\text{switch}}$) primarily determine how much computation we allocate: larger values (and smaller values for $\sigma_{\text{switch}}$) give more correction steps and better quality at higher cost, while smaller values (and larger values for $\sigma_{\text{switch}}$) make the method cheaper but slightly less accurate. In our experiments we chose these values so that FAST-DIPS matches or surpasses the competing methods’ quality while still clearly demonstrating its speed advantage. For users with a stricter compute budget, one can simply reduce $(T,K,S)$ and increase $\sigma_{\text{switch}}$ for latent case (shifting more work into the cheaper pixel regime); the method’s behavior degrades smoothly under such changes.
>
> Finally, we note that baselines such as DAPS, HRDIS, SITCOM, and C-$\Pi$GDM also require hyperparameter selection (number of diffusion steps, data-fidelity weights, inner step sizes, etc.). In all cases we follow the same practice: use authors’ recommended settings or a small coarse validation sweep. The one-time search cost is therefore comparable across methods and is not included for any of them in the reported runtimes. Under these conditions, FAST-DIPS still achieves the reported speedups, so its practical runtime advantage remains intact.

---

> ### Author Response · Authors · 2025-11-22
> **Author response (3/3)**
>
> > W3. Moreover, I am skeptical about the method's scalability to higher resolutions. The computational cost of the VJP/JVP, especially for the latent variant, which requires backpropagation through the decoder, was manageable for the $ 256 \times 256 $ experiments. However, this computation and memory overhead is likely to become a significant bottleneck as resolution increases, potentially limiting the framework's applicability to large-scale problems.
>
> |Method| Res.|Total (s)|Latent corr. (s)|Forward (s)|VJP (s)|JVP (s)|Peak VRAM (GiB)|
> |-|-|-|-|-|-|-|-|
> | Latent DAPS| $256^2$ |84.97|-- |--|--|--|4.40|
> | Latent DAPS| $512^2$ |560.80|--|--|--|--|11.28|
> | Latent-FAST-DIPS (ours)|$256^2$ |38.38| 36.73|13.52| 5.22| 17.31|4.41|
> | Latent-FAST-DIPS (ours)|$512^2$ |243.61| 237.48|90.25|37.04|108.42|11.32|
>
> *Table 2* Scalability of latent solvers on FFHQ at $256^2$ and $512^2$ (10 samples, batch size 1).
>
> We agree that, in the latent variant, differentiating through the decoder via VJP/JVP could in principle become a bottleneck at higher resolutions. To assess this, we instrumented the latent solver and measured runtime and memory at both $256\times256$ and $512\times512$ for 10 reconstructions (batch size 1), and for FAST-DIPS we further decomposed the latent correction time into forward, VJP, and JVP components.
>
> **Memory.**
> At $512^2$, Latent FAST-DIPS uses 11.32\,GiB of peak VRAM versus 11.28\,GiB for Latent DAPS, i.e., a difference of about 0.04\,GiB ($\approx 0.3\%$). In practice, memory is dominated by decoder/U-Net feature maps (shared by all methods), so the extra VJP/JVP graph adds negligible overhead.
>
> **Runtime and operations.**
> From $256^2$ to $512^2$ (a $4\times$ increase in pixels), total runtime grows by $\approx 6.6\times$ for Latent DAPS (84.97s $\to$ 560.80s) and by $\approx 6.3\times$ for Latent FAST-DIPS (38.38s $\to$ 243.61s), while FAST-DIPS remains about $2.2\text{--}2.3\times$ faster at both resolutions. For Latent FAST-DIPS, the latent correction time is split in a stable proportion between forward evaluation of $z \mapsto \mathcal{A}(\mathcal{D}(z))$, VJP (reverse-mode backprop of a scalar data term to the latent), and JVP (forward-mode directional derivative of $\mathcal{A}\circ\mathcal{D}$): at $256^2$, roughly $37\%/14\%/47\%$ of correction time is spent in forward/VJP/JVP, and at $512^2$ these become $\approx 38\%/16\%/46\%$. For convolutional decoders and convolutional/FFT-based $\mathcal A$, these primitives all have comparable asymptotic cost (linear or $N\log N$ in the number of pixels); their different wall-clock times at a fixed resolution arise from constant factors and implementation details (e.g., forward-mode jvp replays parts of the forward with extra bookkeeping, whereas reverse-mode can reuse saved activations and only needs gradients w.r.t. the low-dimensional latent). Crucially, when we move from $256^2$ to $512^2$, the forward, VJP, and JVP times all grow by similar factors ($\approx 6\text{--}7\times$), indicating no super-linear explosion in the gradient terms.
>
> Overall, these measurements show that (i) VJP/JVP through the decoder does not introduce a pathological compute or memory bottleneck beyond the natural cost of high-resolution decoding, and (ii) even at $512^2$, Latent FAST-DIPS retains a substantial $2.2\text{--}2.3\times$ runtime advantage over Latent DAPS at nearly identical peak VRAM.
>
> We hope our response adequately addresses your concerns. We are happy to provide further clarifications if needed.

---

### Author Response · Authors · 2025-11-22
**Common response**

We thank all reviewers for their constructive feedback and summarise below how the revision addresses the main concerns.

**New experiments and broader evaluation**

To address questions about generality and scalability, we added two sets of experiments.
First, besides FFHQ we now evaluate eight inverse problems on a 256×256 ImageNet subset. Table 1 in paper reports FFHQ and ImageNet side by side and shows that FAST-DIPS maintains or improves reconstruction quality while keeping its runtime advantage.
Second, we added a 512×512 Gaussian deblurring experiment in the latent setting on FFHQ, where latent FAST-DIPS improves reconstruction metrics over latent DAPS while remaining about 2.2–2.3× faster at similar peak VRAM.
Finally, we added a JPEG restoration experiment where measurements come from a non-differentiable codec but the forward operator inside FAST-DIPS is a differentiable JPEG surrogate; the qualitative figures show that this surrogate is effective.

**Adjoint-free design and assumptions on $\mathcal{A}$**

We clarify when the adjoint-free formulation is preferable and how costly automatic differentiation through $\mathcal{A}$ is. Users only implement the forward map $\mathcal{A}(\mathbf{x})$ and vector-Jacobian products are obtained from standard autodiff. Avoiding hand-coded adjoints is not just an engineering convenience: it extends applicability to complex nonlinear operators where analytic adjoints are hard or impossible to derive, so any differentiable $\mathcal{A}$ can be plugged in via its VJP/JVP interface. While autodiff introduces a per-step overhead, it enables our analytic VJP/JVP-based step size, which substantially reduces the number of sampling steps. As our 512$^2$ latent study shows, the resulting solver is still $\approx 2.3\times$ faster end-to-end than latent DAPS at similar peak VRAM, and the VJP cost remains within a small constant factor of a manually implemented adjoint. For genuinely non-differentiable degradations we discuss replacing hard operators (e.g., JPEG) with differentiable surrogates, as in the new JPEG experiment.

**Hyperparameters and removal of $\lambda$**

Concerns about hyperparameter tuning led us to simplify the parameterization. In the original submission the trust-region scale involved an extra factor $\lambda$; in the revised paper we absorb $\lambda$ into the ADMM penalty and fix the diffusion trust region as $\gamma_t = \sigma_t^2$. This removes an opaque degree of freedom and leaves the ADMM penalty $\rho$ and the hard-constraint radius $\varepsilon$ as the only method-specific hyperparameters; the remaining parameters $(T,K,S,\sigma_{\text{switch}})$ simply control how much computation one is willing to spend. Appendix Table 6 shows FAST-DIPS to be insensitive to both $\rho$ and $\varepsilon$: sweeping $\varepsilon$ across more than two orders of magnitude (including $\varepsilon=0$) yields essentially unchanged PSNR/LPIPS.

**Implementation update (JVP interface)**

During the discussion, we identified that our initial implementation relied on the legacy `torch.autograd.functional.jvp`, which incurs significant graph construction overhead. We transitioned to the modern `torch.func.jvp` interface, resulting in a substantial speedup with identical numerical behavior. All results reported in the revised paper have been updated using this more efficient implementation.

**Method overview, intuition, and notation**

To mitigate the perceived density of the Method section, we restructured its opening. A new subsection ``High-level overview and design choices'' now explains in plain language what practical issues FAST-DIPS aims to address, how the per-step update is organized, and which theoretical guarantees support these choices. Immediately afterwards, we added a short paragraph that lists what is evaluated at each step so that readers can understand the algorithmic flow before the derivations. We also clarified the role of the local linearization $\mathcal{A}(x-\alpha g)\approx \mathcal{A}(x)-\alpha J_\mathcal{A}(x)g$, explicitly distinguishing the exact linear case from the first-order nonlinear case and pointing to Proposition 3 for the resulting descent guarantee.

**Tables, baselines, and reproducibility**

During our re-evaluation of the baselines, we identified an implementation issue in the official ReSample repository: the normalization step `x = (x + 1.0) / 2.0` was omitted in the forward operators for Phase Retrieval and Nonlinear Deblurring. We have corrected this omission, and updated the scores in the revised table to ensure a strictly accurate and fair comparison.

We reformatted the main quantitative table to respect ICLR page-width constraints and to accommodate the new ImageNet. We also now state explicitly that all FFHQ results in Table 1 of the paper are evaluated on our own validation split, which differs from those used in the original DAPS and SITCOM papers and explains the small numerical discrepancies.

---

### Meta-Review · Area_Chair_nUVC · 2026-01-07

**Summary:**

This paper introduces a diffusion-prior inverse solver with a hard-constrained likelihood correction at each diffusion step. The key point is using an adjoint-free ADMM formulation with an analytic, non-iterative step size computed from a VJP and JVP, eliminating the need for hand-coded adjoints, inner MCMC loops, or iterative optimization, so as to accelerate the sampling process. The approach also extends to latent diffusion priors. Experiments across various tasks show that the approach gets good reconstruction quality while offering faster runtimes than other baselines.

The proposed adjoint-free ADMM design is interesting and looks effective to eliminate hand-coded adjoints by leveraging the standard automatic differentiation, yielding promising speedups while maintaining or improving reconstruction quality. The method is supported by theoretical guarantees as well as thorough empirical results demonstrating good performance compared with other state-of-the-art approaches.

The reviewers raise the questions and concerns on the potential limitation from the reliance on differentiable forward operators,  for example, how to extend to non-differentiable degradations such as JPEG compression, and the computational overhead from autodiff through the forward model (and decoder in latent mode), especially in the case of higher resolutions. The other questions about task-specific hyperparameter tuning, and limited datasets in the experimental evaluation, and the justification for nonlinear problem approximation.

During the rebuttal, the author provides extensive responses with new experimental results to answer these questions including evaluation on ImageNet dataset, hyperparameter tuning simplification, non-differentiable degradations and a revised manuscript in detail. Unfortunately, only one out of four reviewers responded to the rebuttal while the other three reviewers do provide informative and high-quality review comments. From my point of view, most of the concerns have been well addressed with the provided convincing results and analysis and the revised paper is largely improved to reach the bar of publication. Overall, I would be in support of accepting this paper for the interesting exploration and investigation for accelerating the diffusion inverse solvers, which may bring benefits for many practical applications.

**Reviewer Concerns:**

The reviewers raise the questions and concerns on the potential limitation from the reliance on differentiable forward operators,  for example, how to extend to non-differentiable degradations such as JPEG compression, and the computational overhead from autodiff through the forward model (and decoder in latent mode), especially in the case of higher resolutions. The other questions about task-specific hyperparameter tuning, and limited datasets in the experimental evaluation, and the justification for nonlinear problem approximation.

During the rebuttal, the author provides extensive responses with new experimental results to answer these questions including evaluation on ImageNet dataset, hyperparameter tuning simplification, non-differentiable degradations and a revised manuscript in detail. Unfortunately, only one out of four reviewers responded to the rebuttal while the other three reviewers do provide informative and high-quality review comments. From my point of view, most of the concerns have been well addressed with the provided convincing results and analysis and the revised paper is largely improved to reach the bar of publication.

**Reviewer Scores:**

The original scores are mostly borderline leaning towards acceptance. After the rebuttal, I think the detailed response from the author with new experiments and results are helpful to further improve the manuscript. Thus, there could be a chance for reviewers to raise the score.

---

### Decision · Program_Chairs · 2026-01-26

Accept (Poster)